# The MOM1 complex recruits the RdDM machinery via MORC6 to establish de novo DNA methylation

Zheng Li [1,7], Ming Wang [1,7], Zhenhui Zhong [1,7], Javier Gallego-Bartolomé[1,5], Suhua Feng[1,2], Yasaman Jami-Alahmadi[3], Xinyi Wang[1], James Wohlschlegel[3], Sylvain Bischof[1,6], Jeff A. Long[1] & Steven E. Jacobsen [1,2,4] ✉

MORPHEUS' MOLECULE1 (MOM1) is an *Arabidopsis* factor previously shown to mediate transcriptional silencing independent of major DNA methylation changes. Here we find that MOM1 localizes with sites of RNA-directed DNA methylation (RdDM). Tethering MOM1 with an artificial zinc finger to an unmethylated *FWA* promoter leads to establishment of DNA methylation and *FWA* silencing. This process is blocked by mutations in components of the Pol V arm of the RdDM machinery, as well as by mutation of *MICRORCHIDIA 6 (MORC6)*. We find that at some endogenous RdDM sites, MOM1 is required to maintain DNA methylation and a closed chromatin state. In addition, efficient silencing of newly introduced *FWA* transgenes is impaired in the *mom1* mutant. In addition to RdDM sites, we identify a group of MOM1 peaks at active chromatin near genes that colocalized with MORC6. These findings demonstrate a multifaceted role of MOM1 in genome regulation.

Transcriptional silencing is critical to keep transposable elements (TEs) and DNA repeats under control in eukaryotic genomes. The process of transcriptional silencing involves several elaborate mechanisms involving many proteins as well as DNA methylation and histone modifications[1,2]. In *Arabidopsis*, the *MORPHEUS' MOLECULE1* (*MOM1*) gene, which was originally identified with the phenotype of reactivation of a DNA-methylated and silenced hygromycin-resistance transgene in the *mom1* mutant[3], is a distinct component of the transcriptional silencing machinery. In the *mom1* mutant, a set of TEs, mainly located in pericentromeric regions[4,5], is robustly activated without major alteration in DNA methylation patterns[5–7]. In addition, no obvious visible decompaction of heterochromatin at chromocenters was observed in the *mom1* mutant[8–10]. The mechanism of MOM1 mediated silencing remains elusive.

*MOM1* encodes a large protein (2001 amino acids) with sequence homology to the ATPase domain of SWI2/SNF2 family proteins[3].

However, this SNF2 homology sequence is largely dispensable for MOM1's silencing function[11]. Instead, the Conserved MOM1 Motif 2 (CMM2) domain, which is conserved among MOM1 orthologs, is required for the silencing function of MOM1[11]. The CMM2 domain of MOM1 multimerizes with itself and interacts with two PIAS (PROTEIN INHIBITOR OF ACTIVATED STAT)-type SUMO E3 ligase-like proteins, PIAL1 and PIAL2[5,12]. The *pial1 pial2* double mutant phenotype highly resembles the endogenous TE de-repression phenotype of *mom1*[5], suggesting that the PIAL proteins and the MOM1 protein function in the same pathway. However, evidence suggests that the SUMO ligase activity is not required for the transcriptional silencing by PIAL2, and the interaction of MOM1 and PIAL2 with SUMO is also not required for the silencing function of the MOM1 complex[5,13].

RNA directed DNA Methylation (RdDM) is a plant specific pathway responsible for de novo DNA methylation[14]. It also assists in maintaining pre-existing DNA methylation patterns together with other DNA

[1]Department of Molecular, Cell and Developmental Biology, University of California, Los Angeles, CA, USA. [2]Eli & Edythe Broad Center of Regenerative Medicine & Stem Cell Research, University of California at Los Angeles, Los Angeles, CA, USA. [3]Department of Biological Chemistry, University of California, Los Angeles, CA, USA. [4]Howard Hughes Medical Institute, University of California, Los Angeles, CA, USA. [5]Present address: Instituto de Biología Molecular y Celular de Plantas (IBMCP), CSIC-Universitat Politècnica de València, Valencia, Spain. [6]Present address: Department of Plant and Microbial Biology, University of Zurich, Zurich, Switzerland. [7]These authors contributed equally: Zheng Li, Ming Wang, Zhenhui Zhong. ✉e-mail: jacobsen@ucla.edu

methylation mechanisms[15]. The RdDM pathway can be divided into two arms. In the RNA POLYMERASE IV (Pol IV) arm of the RdDM pathway, SAWADEE homeodomain homolog 1 (SHH1) and CLASSY (CLSY) proteins recruit Pol IV to target sites marked by H3K9 di-methylation and unmethylated H3K4 to produce precursor single-stranded RNA (ssRNA) of 30-45 nucleotides (nt) in length[16–19]. RNA-directed RNA polymerase 2 (RDR2) then converts these ssRNAs into double-stranded RNAs (dsRNA), which are then processed by Dicer-like 3 (DCL3) into 24 nt siRNA[20–23]. 24 nt siRNA are then loaded into ARGONAUTE proteins AGO4/6/9, which then participate in the RNA POLYMERASE V (Pol V) arm of the RdDM pathway[16,24–26]. The Pol V arm of the RdDM pathway is initiated by SU(VAR)3-9 homolog 2 (SUVH2) and SUVH9 binding to methylated DNA and recruiting the DDR complex composed of the DEFECTIVE IN RNA-DIRECTED DNA METHYLATION 1 (DRD1), DEFECTIVE IN MERISTEM SILENCING3 (DMS3) and RNA-DIRECTED DNA METHYLATION1 (RDM1) proteins[27–30]. Subsequently, Pol V is recruited by the DDR complex and synthesizes non-coding RNAs which serve as scaffolds for the binding of AGO-siRNA duplexes[17,31–33]. The DNA methyltransferase enzyme DOMAINS REARRANGED METHYL-TRANSFERASE 2 (DRM2) is then recruited to methylate target DNA[34].

RNA-seq analysis shows that the majority of up-regulated genes and TEs in the *mom1* mutant and in the *nrpe1* mutant (mutant of the largest subunit of Pol V) do not overlap[5,35]. In addition, some genes are exclusively up-regulated in the *mom1 nrpe1* double mutant[35], and a mutant allele of *nrpe1* was identified in a screen for enhancers of the de-repression of a transgenic luciferase reporter in the *mom1* background[35]. These studies suggest that, although MOM1 mediated transcriptional silencing and RdDM function as two different pathways, they also can act cooperatively to silence some endogenous and transgene targets.

The *Arabidopsis* microrchidia (MORC) proteins were discovered as additional factors required for gene silencing downstream of DNA methylation[36]. In addition, MORCs associate with components of the RdDM pathway, are loaded onto sites of RdDM and are needed for the efficiency of RdDM maintenance at some sites[37–40]. The connection between the RdDM pathway and the MORC proteins has also been demonstrated through experiments targeting the *FWA* gene. In wild type plants, *FWA* expression is silenced in all tissues except the endosperm due to DNA methylation in the promoter[41]. In the *fwa-4* epi-mutant (*fwa*), the *FWA* gene promoter is unmethylated leading to constitutive expression of the *FWA* gene and late flowering phenotype[42]. Tethering MORC proteins to the unmethylated promoter of the *FWA* gene in the *fwa* mutant via protein fusion to an artificial zinc finger protein 108 (ZF) led to efficient methylation of the promoter via recruitment of the RdDM machinery[40,43]. In addition, mutations of the *MORC* genes impair the efficient de novo methylation and silencing of *FWA* transgenes[40].

Previous studies have identified functional similarities between MORC proteins and the MOM1 complex. Multiple screens using silenced transgene reporters have identified mutations in both *MOM1* and *MORC6*[5,6], suggesting that they are both required for maintaining the silenced state of these transgenes. Analysis of gene expression defects in mutants has shown that most of derepressed TEs in the *morc6* mutant were also derepressed in *mom1*, while another group of TEs are uniquely derepressed only in the *mom1 morc6* double mutant[6]. Thus, investigating the relationship between the RdDM machinery, MORC proteins and the MOM1 complex should help to understand the convergence and divergence in their functions.

In this study, by performing chromatin immunoprecipitation sequencing (ChIP-seq), we observe a strong colocalization of MOM1 complex components, with the MORC6 protein and RdDM sites. Tethering of MOM1 complex components to the *FWA* promoter in the *fwa* mutant by ZF fusion leads to the establishment of DNA methylation and silencing of the *FWA* gene. By transforming ZF fusions into mutants, we discover that the establishment of DNA methylation by ZF-MOM1 is not only blocked by the mutants of the downstream components of the RdDM pathway, but also blocked in *morc6*. Furthermore, an interaction between PIAL2 and MORC6 is detected by a yeast two-hybrid (Y2H) assay and co-immunoprecipitation (co-IP). In addition, efficient de novo methylation and silencing of an *FWA* transgene are impaired in the *mom1* and the *pial1/2* mutants. Consistent with the divergent function of the MOM1 complex and the RdDM pathway, the MOM1 complex is more enriched at TEs in pericentromeric regions, while Pol V is more enriched at TEs in the chromosome arms. MOM1 also binds to a group of RdDM independent sites, at active, unmethylated, and accessible chromatin. These results highlight the functions for MOM1 in genome regulation and help to clarify the relationship between MOM1, MORCs and RdDM.

## Results

### MOM1 complex colocalizes with RdDM sites

Previously, it was shown that MOM1, PIAL1 and PIAL2 form a high molecular weight complex in vivo[5]. In addition, MOM1 Immunoprecipitation-Mass Spectrometry (IP-MS) pulled down other interactors such as AIPP3 and PHD1[5]. To comprehensively identify interacting components of the MOM1 complex, we repeated the IP-MS experiments of MOM1 protein with a 3X-FLAG epitope tag and observed that, consistent with previous reports, PIAL1, PIAL2, PHD1 and AIPP3 were pulled down (Table 1 and Supplementary Data 1). In addition, the MOM2 protein, which was predicted to be a non-functional homolog of MOM1, was identified in the MOM1 IP-MS (Table 1 and Supplementary Data 1). Previous IP-MS of the AIPP3 protein pulled down other protein components such as PHD2 (also called PAIPP2), PHD3 (also called AIPP2) and CPL2, in addition to PHD1[44–46].

**Table 1 | IP-MS of MOM1, MOM2, PIAL2, PHD1, and AIPP3**

| Gene | Protein | Col-0 | | MOM1-FLAG | | MOM2-FLAG | | | PIAL2-FLAG | | | PHD1-FLAG | | AIPP3-FLAG | |
|---|---|---|---|---|---|---|---|---|---|---|---|---|---|---|---|
| | | Rep1 | Rep2 | Rep1 | Rep2 | Rep1 | Rep2 | Rep3 | Rep1 | Rep2 | Rep3 | Rep1 | Rep2 | Rep1 | Rep2 |
| AT1G08060 | MOM1 | 0 | 1 | 607 | 789 | 51 | 317 | 204 | 244 | 204 | 225 | 158 | 135 | 124 | 121 |
| AT2G28240 | MOM2 | 0 | 0 | 76 | 80 | 53 | 529 | 912 | 62 | 38 | 48 | 9 | 2 | 3 | 1 |
| AT1G08910 | PIAL1 | 0 | 0 | 21 | 31 | 3 | 29 | 13 | 10 | 4 | 2 | 3 | 1 | 4 | 2 |
| AT5G41580 | PIAL2 | 1 | 3 | 146 | 162 | 21 | 137 | 102 | 241 | 263 | 328 | 21 | 12 | 15 | 13 |
| AT1G43770 | PHD1 | 0 | 0 | 24 | 40 | 0 | 44 | 22 | 34 | 31 | 29 | 190 | 145 | 62 | 65 |
| AT4G11560 | AIPP3 | 0 | 0 | 68 | 87 | 10 | 105 | 43 | 95 | 52 | 75 | 224 | 143 | 1412 | 1313 |
| AT5G01270 | CPL2 | 0 | 0 | 0 | 0 | 0 | 0 | 0 | 0 | 0 | 0 | 0 | 0 | 16 | 15 |
| AT5G16680 | PHD2 | 0 | 0 | 0 | 0 | 0 | 0 | 0 | 0 | 0 | 0 | 0 | 0 | 212 | 214 |
| AT3G02890 | PHD3 | 0 | 0 | 0 | 0 | 0 | 0 | 0 | 0 | 0 | 0 | 0 | 0 | 126 | 145 |

Col-0 plants and FLAG epitope tagged MOM1, MOM2, PIAL2, PHD1 and AIPP3 transgenic plants were used for IP-MS.
MS/MS counts from MaxQuant output are listed.
Rep represents replicates.

To facilitate the dissection of the interacting components, we performed IP-MS with FLAG tagged MOM2, PIAL2, PHD1 and AIPP3. AIPP3 pulled down MOM1, MOM2, PIAL1, PIAL2, PHD1, as well as CPL2, PHD2 and PHD3 (Table 1 and Supplementary Data 1). MOM2, PIAL2 and PHD1 pulled down each other reciprocally, as well as the PIAL1 and MOM1 protein, but no peptides of CPL2, PHD2 and PHD3 (Table 1 and Supplementary Data 1). Consistent with previous studies showing AIPP3 forms a complex with CPL2, PHD2 and PHD3[44–46], AIPP3 appears to be a component of multiple protein complexes, including the MOM1 protein complex.

To study the function of the MOM1 complex, ChIP-seq was performed in FLAG or Myc tagged MOM1, PIAL2, PHD1 and AIPP3 transgenic lines. Surprisingly, MOM1, PHD1, AIPP3, and PIAL2 were all highly colocalized with Pol V at RdDM sites (Fig. 1 a and b). To further validate

colocalization of the MOM1 complex with RdDM sites, we performed crosslinking IP-MS of FLAG tagged MOM1 and observed that in addition to the MOM1 complex components, several proteins in the RdDM machinery, including NRPD2 (subunit of Pol V and Pol IV), NRPE1 (subunit of Pol V), DMS3 and SPT5L ($P = 0.01243$) were also enriched (Fig. 1c and Supplementary Data 2). Interestingly, we also observed a strong enrichment of MORC1 and MORC6 in the MOM1 crosslinking IP-MS (Fig. 1c and Supplementary Data 2), suggesting that the RdDM machinery, the MORC proteins and the MOM1 complex are co-located at the same loci, either because they are crosslinked by co-bound stretches of chromatin, or because the crosslinking process enhanced relatively weak interactions between the proteins.

Further examination of the MOM1 ChIP-seq signal over the AIPP3 peaks suggested that a group of AIPP3 binding loci were not enriched

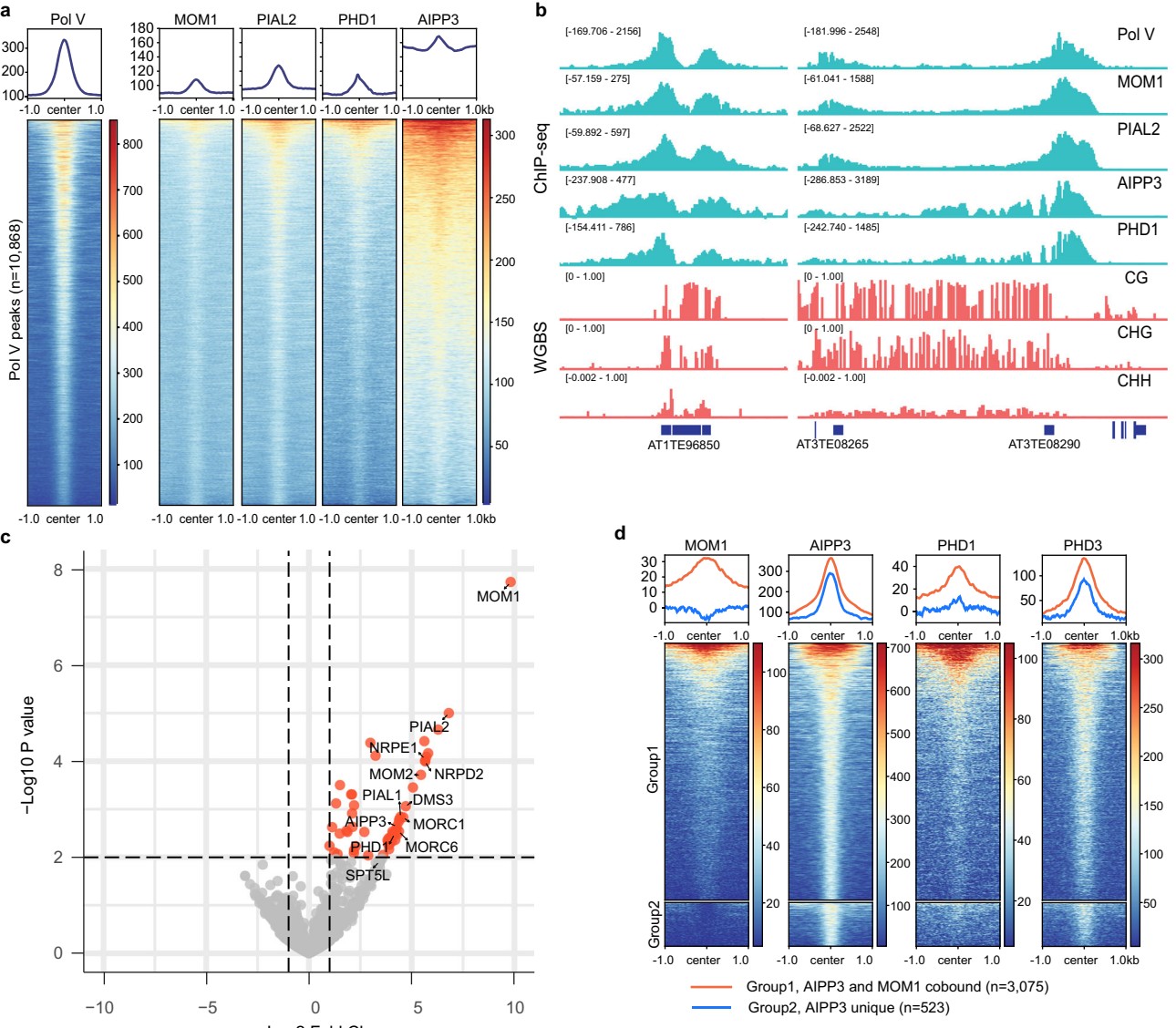

**Fig. 1 | The MOM1 complex colocalizes with RdDM sites. a** Metaplots and heatmaps representing ChIP-seq signals of Pol V, MOM1-Myc, PIAL2-Myc, PHD1-FLAG, and AIPP3-FLAG over Pol V peaks ($n = 10{,}868$). ChIP-seq signal of control samples were subtracted for plotting. **b** Screenshots of Pol V, MOM1-Myc, PIAL2-Myc, AIPP3-FLAG and PHD1-FLAG ChIP-seq signals with control ChIP-seq signal subtracted and CG, CHG, and CHH DNA methylation level by whole genome bisulfite sequencing (WGBS) over representative RdDM sites. **c** Volcano plot showing proteins that have significant interactions with MOM1 as detected by crosslinking IP-MS, with RdDM pathway components and MOM1 complex components labeled. Crosslinking IP-MS

of Col-0 plant tissue was used as control. The empirical Bayes test performed by LIMMA was used for statistical analysis. **d** AIPP3-FLAG ChIP-seq peaks were divided into two groups: Group 1 peaks ($n = 3{,}075$) have MOM1-Myc ChIP-seq signal enriched and Group 2 peaks ($n = 523$) have no enrichment of MOM1-Myc ChIP-seq signal. Metaplots and heatmaps representing ChIP-seq signals of MOM1-Myc, AIPP3-FLAG, PHD1-FLAG and PHD3-FLAG over these two groups of AIPP3 peaks. ChIP-seq signal of control samples were subtracted for plotting. Source data are provided as a Source Data file.

for MOM1 (Fig. 1d). We named the group of AIPP3 peaks that have MOM1 ChIP-seq signal enriched as Group 1 peaks and those with no MOM1 enrichment as Group 2 peaks. Consistent with our IP-MS data suggesting that PHD1 is a MOM1 complex component, PHD1 ChIP-seq signal was predominantly enriched in AIPP3 Group1 peaks which also bound to MOM1 (Fig. 1d and Supplementary Fig. 1). We also performed ChIP-seq with FLAG tagged PHD3 transgenic plants. In contrast to PHD1, PHD3 ChIP-seq signal was enriched in both groups of AIPP3 peaks, closely resembling the pattern of AIPP3 ChIP-seq signal (Fig. 1d and Supplementary Fig. 1). These data further suggests that AIPP3 exists in multiple protein complexes including the MOM1 complex.

**MOM1-ZF triggers DNA methylation at the *FWA* promoter**
Since MOM1 localized at RdDM sites, and ZF fusions of RdDM components have been shown to silence *FWA* expression in the *fwa* mutant[43], we investigated whether tethering the components of the MOM1 complex could also lead to the silencing of *FWA* expression. We created ZF fusion proteins with MOM1, MOM2, PIAL1, PIAL2, AIPP3 and PHD1 and transformed them into the *fwa* mutant. ZF fusion of MOM1, MOM2, PIAL1, PIAL2 and PHD1 restored the early flowering phenotype (Fig. 2a, Supplementary Fig. 2a), repressed *FWA* expression (Fig. 2b),

and induced DNA methylation at the *FWA* promoter region as detected by bisulfite amplicon sequencing analysis (BS-PCR-seq) (Fig. 2c). The DNA methylation induced at the *FWA* promoter region was retained in the transgene-free T2 plants, showing that the newly established DNA methylation was heritable (Fig. 2c). PIAL1-ZF was somewhat less efficient at restoring the early flowering phenotype in the T1 population (Supplementary Fig. 2a). However, reduced *FWA* mRNA levels and increased *FWA* promoter DNA methylation were detected in some PIAL1-ZF T1 plants (Supplementary Fig. 2b), and plants with similar flowering time to the Col-0 were observed from three T2 populations of the earliest flowering T1 plants (Fig. 2a, Supplementary Fig. 2a). In addition, DNA methylation at the *FWA* promoter region was retained in T2 plants free of PIAL1-ZF transgenes, showing that PIAL1-ZF can also induce heritable DNA methylation (Fig. 2c). AIPP3-ZF led to a slightly early flowering time in the T1 population compared to the *fwa* control population, however, zero T1 transgenic plants and very few T2 plants flowered as early as the Col-0 control plants (Supplementary Fig. 2a, c). A low level of DNA methylation in the *FWA* promoter region, mainly methylation in the CHH sequence context, was detected in the AIPP3-ZF T2 plants which were positive for the transgene (Supplementary Fig. 2d). However, no DNA methylation was detected in transgene-free

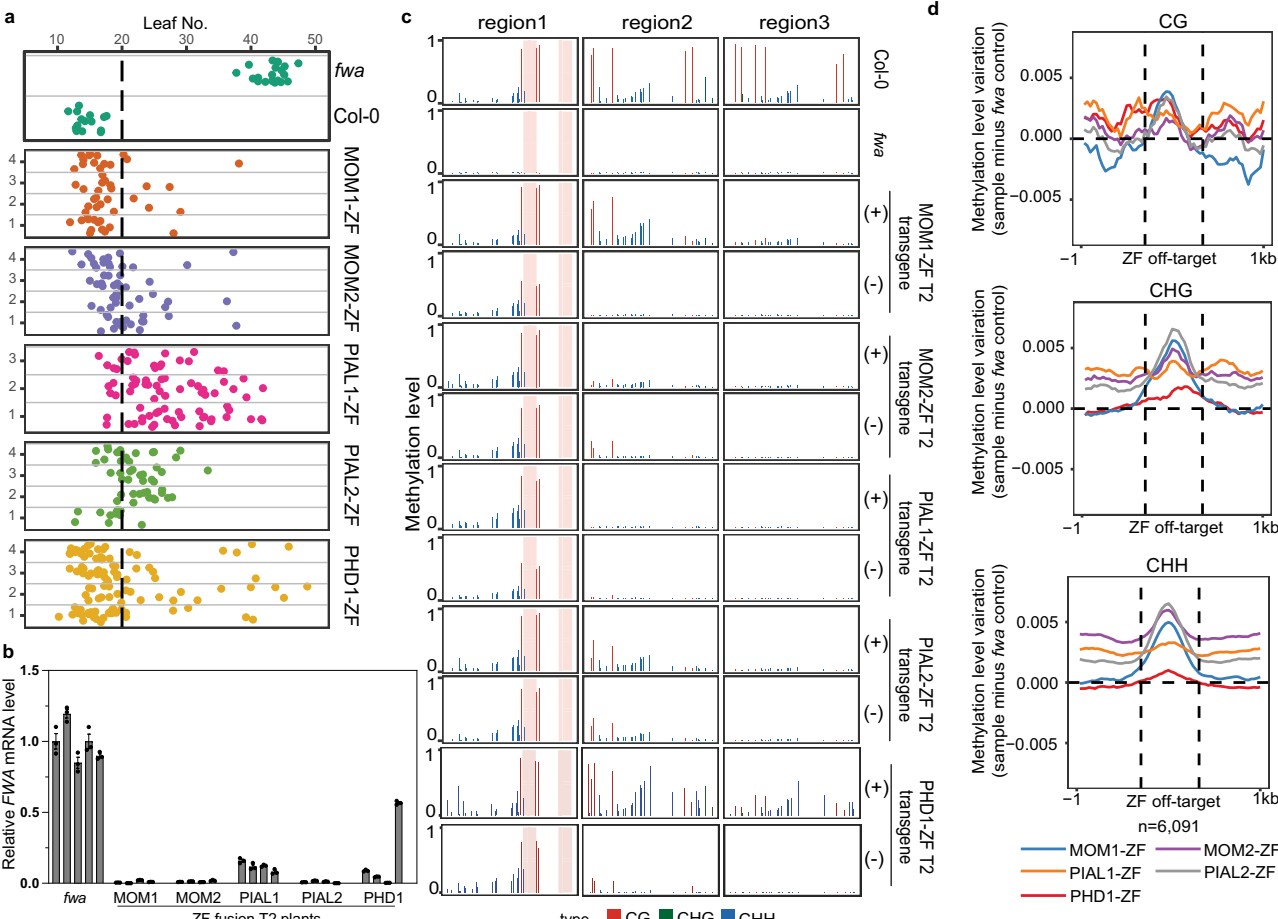

**Fig. 2 | ZF tethering of the MOM1 complex to the *FWA* promoter triggers DNA methylation and *FWA* silencing. a** Flowering time of *fwa*, Col-0 and representative T2 lines of MOM1-ZF, MOM2-ZF, PIAL1-ZF, PIAL2-ZF and PHD1-ZF in the *fwa* background. The numbers of independent plants (*n*) scored for each population and detailed statistics of flowering time comparison between different populations are listed in Supplementary Data 5. **b** qRT-PCR showing the relative mRNA level of *FWA* gene in the leaves of *fwa* plants, and four T2 plants of MOM1-ZF, MOM2-ZF, PIAL1-ZF, PIAL2-ZF and PHD1-ZF in the *fwa* background. Bar plots and error bars indicate the mean and standard error of three technical replicates, respectively, with

individual technical replicates shown as dots. **c** CG, CHG, and CHH DNA methylation levels over *FWA* promoter regions measured by BS-PCR-seq in Col-0, *fwa* and representative T2 plants of MOM1-ZF, MOM2-ZF, PIAL1-ZF, PIAL2-ZF and PHD1-ZF in the *fwa* background with (+) or without (-) corresponding transgenes. Pink vertical boxes indicate ZF binding sites. **d** Metaplots showing relative variations (sample minus *fwa* control) of CG, CHG, and CHH DNA methylation levels over ZF off-target sites in representative T2 plants of MOM1-ZF, MOM2-ZF, PIAL1-ZF, PIAL2-ZF and PHD1-ZF in the *fwa* background measured by WGBS. Source data are provided as a Source Data file.

T2 plants segregating in the same T2 populations (Supplementary Fig. 2d). These data suggests that the establishment of DNA methylation by AIPP3-ZF is much weaker compared to other MOM1 complex components. Previous work reported that, in addition to the designed binding site in the *FWA* promoter, ZF also binds to many off-target sites in the genome[43]. Whole genome bisulfite sequencing (WGBS) showed that MOM1-ZF, MOM2-ZF, PIAL1-ZF, PIAL2-ZF and PHD1-ZF also enhanced DNA methylation at ZF off-target sites (Fig. 2d, Supplementary Fig. 3a, b). Overall, these results show that ZF fusions of the components of the MOM1 complex are able to trigger the establishment of DNA methylation and silence *FWA* expression in the *fwa* mutant, as well as establish methylation at other ZF off-target sites.

The CMM2 domain has been shown to be essential for the transcriptional gene silencing function of the MOM1 protein[11,12]. We found that a ZF fusion with the CMM2 domain together with a nuclear localization signal (called miniMOM1)[11] was efficient at targeting heritable *FWA* methylation (Supplementary Fig. 3c, d). We performed IP-MS with a miniMOM1-FLAG line and found peptides for MOM2, PIAL1, and PIAL2, but not for AIPP3 or PHD1 (Supplementary Data 1). These results suggest that AIPP3 and PHD1 may be dispensable for the targeting of methylation to *FWA* promoter.

To begin to genetically dissect the requirements for MOM1-mediated establishment of *FWA* methylation and silencing, we first transformed MOM1-ZF and PHD1-ZF into *mom1 fwa* and *phd1 fwa* mutant backgrounds (Supplementary Fig. 4a). MOM1-ZF was able to trigger early flowering in *phd1 fwa*, positioning MOM1 downstream of PHD1 (Supplementary Fig. 4a). Consistent with this order of action, the *mom1* mutant blocked PHD1-ZF activity (Supplementary Fig. 4a). PHD1-ZF activity was also blocked in the *aipp3 fwa* mutant (Supplementary Fig. 4a). These results are consistent with IP-MS result showing that the MOM1-PHD1 interaction was abolished in the *aipp3-1* mutant (Supplementary Data 1).

To further dissect the hierarchy of action of MOM1 components, we transformed PIAL2-ZF into *aipp3 fwa*, *phd1 fwa*, *mom2 fwa* and *mom1 fwa* mutant backgrounds and found that PIAL2-ZF triggered an early flowering phenotype in all mutant backgrounds (Supplementary Fig. 4b), suggesting that PIAL2 might act at the most downstream position within the MOM1 complex. However, we also transformed MOM1-ZF into *aipp3 fwa*, *mom2 fwa* and *pial1/2 fwa*, and found that MOM1-ZF was also able to trigger early flowering in all these mutant backgrounds (Supplementary Fig. 4a), suggesting that MOM1 acts at a step parallel with PIAL1/2 in targeting DNA methylation. We did however observe that MOM1-ZF showed a lower efficiency of triggering early flowering in the *pial1/2 fwa* mutant compared to wild type or the other mutants (Supplementary Fig. 2a and 4a), suggesting that PIAL1/2 is required for the full functionality of MOM1-ZF. We also transformed MOM2-ZF into *aipp3 fwa*, *phd1 fwa*, *mom1 fwa*, and *pial1/2 fwa*, and MOM2-ZF was able to trigger early flowering in all the mutants except in the *pial1/2 fwa* background (Supplementary Fig. 4b). As a control, we compared the flowering time in the mutant backgrounds without transgenes. *mom1 fwa* flowered at a similar time as compared to *fwa*, while *pial1/2 fwa* and *aipp3 fwa* flowered slightly earlier (Supplementary Fig. 4c), suggesting that the deficiency in triggering early flowering by ZF fusion proteins in these backgrounds is not due to differences in flowering time of mutant backgrounds. In summary, these results suggest that MOM1, and especially PIAL1/PIAL2 are acting as the most downstream factors in the MOM1 complex for establishing DNA methylation at the *FWA* promoter.

### The MOM1 complex recruits the Pol V arm of the RdDM machinery via MORC6

Because the tethering of RdDM components to *FWA* has been previously shown to efficiently establish methylation of *FWA*[27,43], we hypothesized that MOM1-ZF established *FWA* DNA methylation by recruiting the RdDM machinery. To test this hypothesis, we

transformed PIAL2-ZF and MOM1-ZF into *fwa* backgrounds in which RdDM mutations had been introgressed, including *nrpd1*, *suvh2/9*, *dms3*, *drd1*, *rdm1*, *nrpe1*, and *drm1/2*[43]. PIAL2-ZF and MOM1-ZF were still capable of triggering an early flowering phenotype in *nrpd1 fwa* (the largest subunit of Pol IV), suggesting that Pol IV mediated siRNA biogenesis was not needed for methylation targeting by the MOM1 complex (Fig. 3a). These fusions were also capable of triggering silencing in the *suvh2/9 fwa* mutant background (Fig. 3a), showing that the SUVH2 and SUVH9 factors that normally recruit the DDR complex and Pol V to chromatin were not needed for silencing. However, silencing activity of PIAL2-ZF and MOM1-ZF was blocked by DDR component mutations (*dms3*, *drd1*, and *rdm1*) as well as by mutations in the largest subunit of Pol V (*nrpe1*) and the DRM de novo methyltransferases (*drm1/2*), as no PIAL2-ZF or MOM1-ZF T1 plants showed early flowering in these backgrounds (Fig. 3a). Some of MOM1-ZF T1 plants displayed intermediate flowering time (20–30 true leaves) in *drm1/2 fwa*, *dms3 fwa*, *drd1 fwa* and *rdm1 fwa* backgrounds (Fig. 3a). However, *FWA* gene expression was not decreased in the six MOM1-ZF T1 plants in the *drm1/2 fwa* background which had the earliest flowering time (23–25 true leaves) from this population (Supplementary Fig. 5), suggesting that the intermediate flowering phenotype is likely due to other factors such as plant stress rather than *FWA* silencing. Overall, these results place the action of PIAL2-ZF and MOM1-ZF upstream of the DDR complex.

Interestingly, it was previously shown that MORC6-ZF showed an identical pattern of triggering *FWA* methylation in wild type, *nrpd1*, and *suvh2/9*, but not in *dms3*, *drd1*, *rdm1*, *nrpe1*, or *drm1/2*[43]. This similarity prompted us to test the targeting of PIAL2-ZF, MOM1-ZF, MOM2-ZF, PIAL1-ZF and PHD1-ZF in the *morc6 fwa* genetic background. Interestingly, we found that all these ZF fusions failed to trigger *FWA* silencing in *morc6 fwa* (Fig. 3a and Supplementary Fig. 4d), suggesting that the MOM1 complex acts upstream of MORC6. To further confirm this order of action we transformed MORC6-ZF into *fwa* backgrounds in which the *mom1-3*, *mom2-1*, *pial1/2*, *phd1-2* and *aipp3-1* mutants had been introgressed. We found that MORC6-ZF could successfully target silencing of *FWA* in all these backgrounds (Supplementary Fig. 4d), confirming that MORC6 acts downstream of the MOM1 complex in the targeting of *FWA* silencing. We also performed ChIP-seq of Myc-tagged MORC6 in the *morc6-3* mutant background. Similar to the MOM1 complex reported here, and MORC4 and MORC7 proteins reported previously[40], we observed that MORC6 was highly colocalized with Pol V at RdDM sites (Fig. 3b, c).

Given that PIAL1/PIAL2 and MOM1 appeared to be the most downstream critical components of the MOM1 complex required for triggering *FWA* methylation, and that ZF fusions of these proteins failed to trigger methylation in a *morc6* mutant, we reasoned at least one of these components might physically interact with MORC6. Indeed, we found that PIAL2 was able to interact with MORC6 in a Yeast Two-Hybrid assay (Fig. 3d). We also confirmed this interaction by an in vivo co-immunoprecipitation assay, observing that MORC6-FLAG was able to interact with PIAL2-Myc (Fig. 3e). While there could certainly be other important interactions, these results suggest that the MOM1 complex likely recruits MORC6 in part via a physical interaction between PIAL2 and MORC6. MORC6 then triggers *FWA* methylation via its interaction with the RdDM machinery as previously reported[40].

To investigate if the MOM1 complex also recruits the MORC6 protein at other loci, ChIP-seq was performed with Myc-tagged MORC6 in the backgrounds of Col-0, *morc6-3 mutant*, *mom1-3* mutant and *pial1/2* double mutant. MORC6 ChIP-seq signal over Pol V peaks was strongly decreased in the *mom1-3* and *pial1/2* mutant backgrounds compared to that in the backgrounds of Col-0 and *morc6-3* mutant (Supplementary Fig. 6a and b), while the MORC6-Myc protein expression levels were not decreased in the *mom1-3* and *pial1/2* mutant backgrounds (Supplementary Fig. 6c). At the same time, there was still residue MORC6 ChIP-seq signal over Pol V peaks in the *mom1-3*

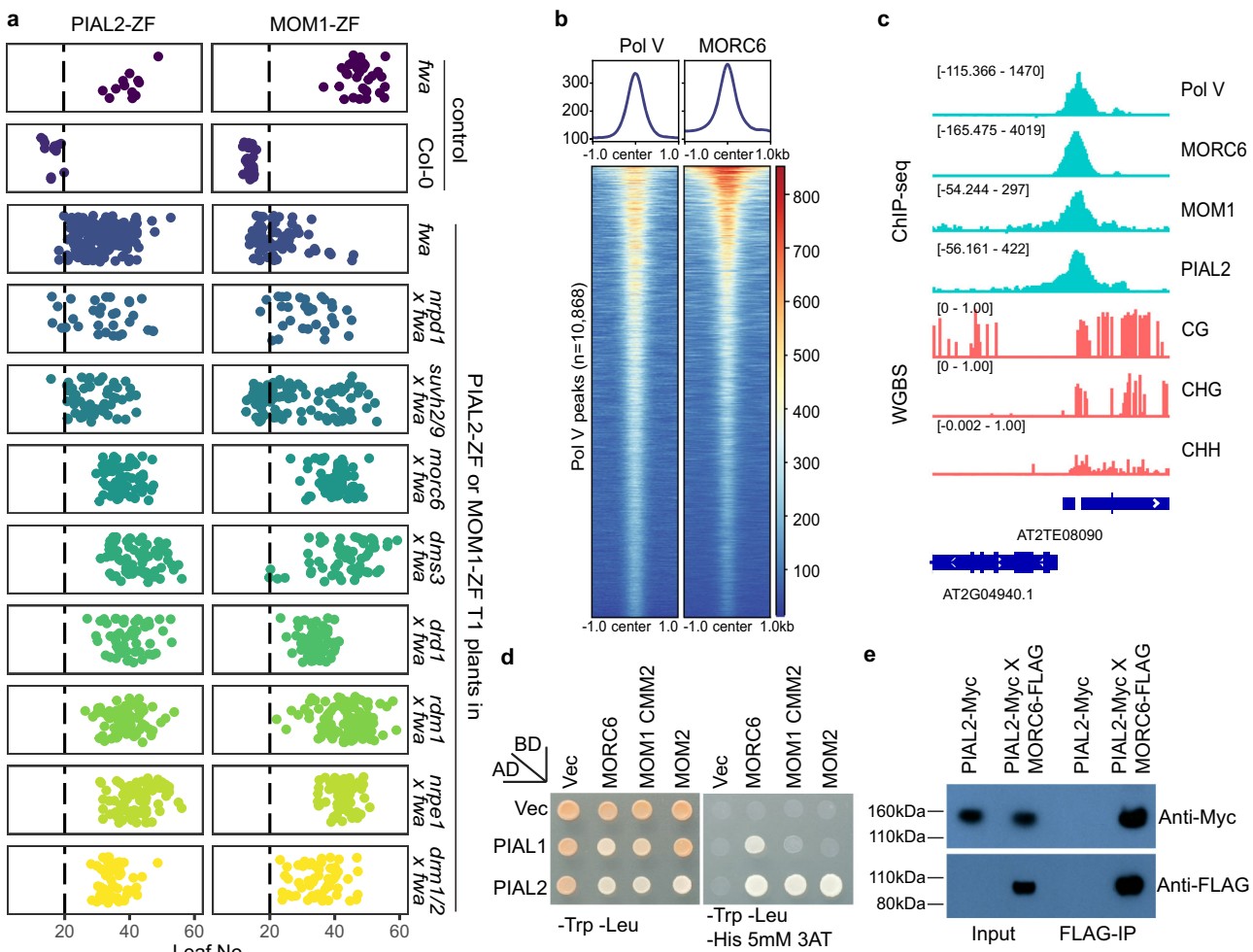

**Fig. 3 | MOM1-ZF recruits the Pol V arm of the RdDM machinery via MORC6.**
**a** Flowering time of *fwa*, Col-0, and T1 lines of PIAL2-ZF and MOM1-ZF in the *fwa* mutant backgrounds as well as in backgrounds of *fwa* introgressed mutants, including *nrpd1, suvh2/9, morc6, dms3, drd1, rdm1, nrpe1* and *drm1/2*. The numbers of independent plants (*n*) scored for each population and detailed statistics of flowering time comparison between different populations are listed in Supplementary Data 5. **b** Metaplots and heatmaps representing ChIP-seq signals of Pol V and MORC6-Myc over Pol V peaks (*n* = 10,868). ChIP-seq signal of control samples were subtracted for plotting. **c** Screenshots of Pol V, MORC6-Myc, MOM1-Myc and

PIAL2-Myc ChIP-seq signals with control ChIP-seq signals subtracted and CG, CHG, and CHH DNA methylation level by WGBS over a representative RdDM site. **d** Yeast Two-Hybrid assay showing in vitro direct interactions between PIAL1 and PIAL2 with MORC6 and the MOM1 CMM2 domain, as well as between PIAL2 and MOM2. This experiment was repeated twice independently with similar results. **e** PIAL2 and MORC6 in vivo interaction shown by co-immunoprecipitation (Co-IP) in MORC6-FLAG and PIAL2-Myc crossed lines. This experiment was repeated twice independently with similar results. Source data are provided as a Source Data file.

and pial1/2 mutant backgrounds (Supplementary Fig. 6a, b). These data suggest that MORC6 is recruited to RdDM sites by the MOM1 complex as well as other mechanisms.

## The MOM1 complex facilitates the process of transgene silencing

Several previous screens identified MOM1 as a key component in the maintenance of the silenced state of the transgene reporters used in the screen[3,6]. RdDM is involved in the maintenance of DNA methylation, but also in the initial establishment of methylation. For example, studies have shown that when an extra copy of the *FWA* gene is introduced into *Arabidopsis* plants via *Agrobacterium*-mediated transformation, it is very efficiently methylated and silenced in the wild type background. However, this methylation and silencing is blocked in RdDM mutants, leading to overexpression and a late flowering phenotype[14,28,47]. Interestingly, the silencing of *FWA* transgenes was previously shown to be less efficient in the *morc* mutants[40]. Since the MOM1 complex is closely linked with the RdDM machinery and MORC6, we suspected that the MOM1 complex may also facilitate the efficient establishment of transgene silencing. To test this, the *FWA*

transgene was transformed into Col-0 plants (wild type) and the mutant background of *nrpe1-11, mom1-3, pial1/2, mom2-22, aipp3-1* and *phd1-2*. As expected[40], the T1 transgenic plants in the *nrpe1-11* background flowered much later (mean leaf number: 33.81) compared to those in the wild type background (mean leaf number: 15.91) (Fig. 4a and Supplementary Fig. 7a). We found that T1 plants containing the *FWA* transgene in *mom1-3* or *pial1/2* mutant backgrounds also flowered later than in those in the Col-0 background, with a mean leaf number of 27.55 (*mom1*) and 31.98 (*pial1/2*) (Fig. 4a and Supplementary Fig. 7a). We examined four late flowering T1 plants in each of the *mom1-3* and *pial1/2* mutant backgrounds and observed that, consistent with their late flowering phenotype, *FWA* mRNA levels were higher than in the wild type background (Fig. 4b upper panel). The unmethylated *FWA* promoter DNA fraction, as detected with McrBC digestion assay, was also higher in these T1 plants compared to wild type, suggesting that efficient establishment of DNA methylation on the *FWA* transgene was impaired in *mom1-3* and *pial1/2* mutants (Fig. 4b lower panel).

Although a small number of T1 *FWA* transgenic plants with a late flowering time was also observed in the *mom2-2, aipp3-1* and *phd1-2* backgrounds, the average flowering time of these T1 plants was not

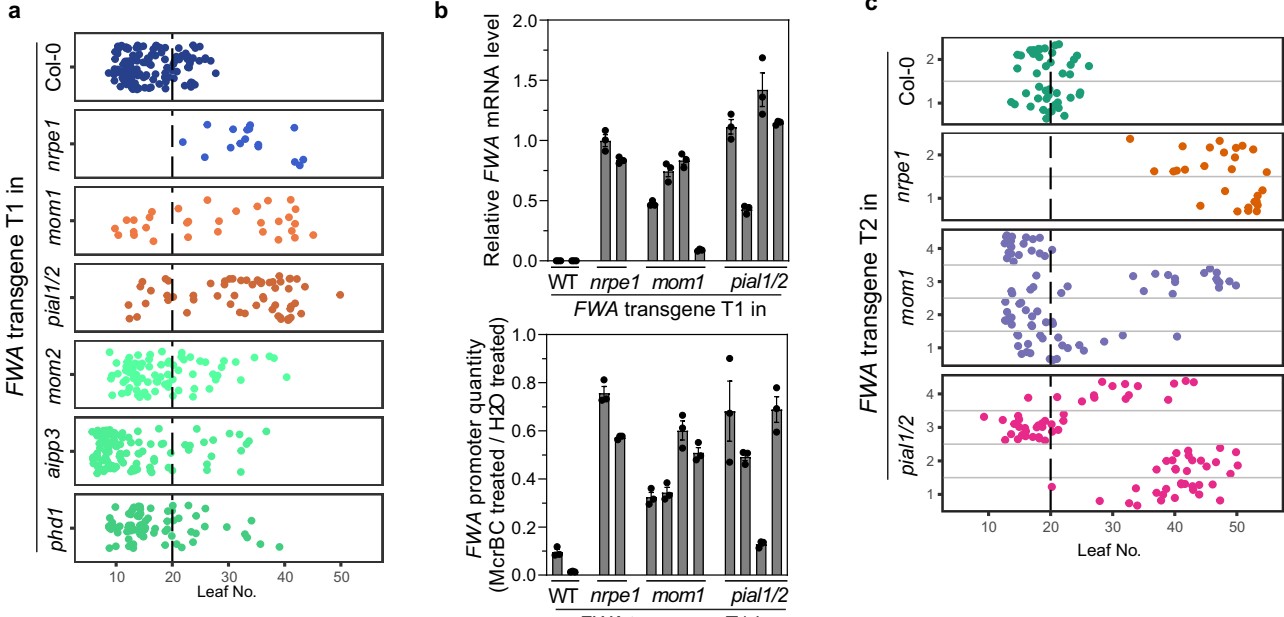

**Fig. 4 | The MOM1 complex facilitates the process of transgene silencing.**
**a** Flowering time of *FWA* transgene T1 plants in the Col-0, *nrpe1-11*, *mom1-3*, *pial1/2*, *mom2-2*, *aipp3-1* and *phd1-2* genetic backgrounds. **b** Relative *FWA* mRNA level (upper panel) and relative *FWA* promoter DNA quantity after McrBC treatment (lower panel) of four late-flowering *FWA* transgene containing T1 plants in the *mom1-3* and *pial1/2* genetic backgrounds. *FWA* transgene containing T1 plants in the Col-0 and *nrpe1-11* backgrounds were used as controls. Bar plots and error bars indicate the mean and standard error of three technical replicates, respectively, with individual technical replicates shown as dots. **c** Flowering time of *FWA* transgene T2 plants in the Col-0, *nrpe1-11*, *mom1-3* and *pial1/2* genetic backgrounds. For **a**, **c**, the numbers of independent plants (*n*) scored for each population and detailed statistics of flowering time comparison between different populations are listed in Supplementary Data 5. Source data are provided as a Source Data file.

significantly later than that of the T1 plants in the wild type background (Fig. 4a and Supplementary Fig. 7a). In fact, the *FWA* transgene T1 population in the *aipp3-1* background flowered significantly earlier than in wild type (Supplementary Fig. 7a), likely due to the fact that the *aipp3-1* mutant itself flowers earlier than wild type plants (Supplementary Fig. 7b), as previously reported[46]. These data suggests that MOM2, AIPP3 and PHD1 contribute minimally to efficient silencing of the *FWA* transgene, whereas MOM1 and PIAL1/2 contribute significantly.

In strong RdDM mutants such as *nrpe1*, the *FWA* transgene stays unmethylated and all of the T2 offspring plants with the *FWA* transgene show a late flowering phenotype[40]. We grew the T2 populations of four late flowering T1 plants in each of the *mom1-3* and *pial1/2* backgrounds and scored for their flowering time. In T2 plant populations in *mom1-3* line 2 and line 4, as well as in *pial1/2* line 3, all transgene positive plants showed a relatively early flowering time, similar to controls of T2 plants with *FWA* transgene in Col-0 background (Fig. 4c). However, in the other T2 populations tested, we observed transgene positive plants with flowering time spanning from very late to early (*mom1-3* T2 line 1 and line3, in *pial1/2* T2 line 1 and line 4), as well as one line with 100% late flowering plants (*FWA* transgene in *pial1/2* line 2) (Fig. 4c). These data suggests that instead of completely blocking *FWA* transgene silencing as in strong RdDM mutants like *nrpe1*, mutation of *MOM1* or *PIAL1/2* reduces the efficiency of *FWA* transgene silencing, similar to what was previously observed for mutation of *MORC* genes[40].

## MOM1 influences DNA methylation and chromatin accessibility at some RdDM sites

The strong co-localization of the MOM1 complex with RdDM sites suggests that the MOM1 complex might facilitate the endogenous function of the RdDM machinery. To test this hypothesis, we performed Whole Genome Bisulfite Sequencing (WGBS) in *mom1-3*, *pial1/2*, *phd1-2*, *phd1-3*, *aipp3-1*, and *mom2-2* and analyzed these together

with previously published WGBS data from the *morc6-3*[23] and *morc1/4/5/6/7* hextuple (*morchex*)[39] mutants, followed by analysis using the High-Confidence Differentially Methylated Regions (hcDMRs) pipeline[7]. We observed 120 hypo CHH hcDMRs in *mom1-3* (shared by two replicates), and 93 hypo CHH hcDMRs in the *pial1/2* double mutant. Over these hypo CHH hcDMRs, Pol V ChIP-seq signal was enriched, and the CHH methylation level were strongly decreased in the *nrpe1* mutant, suggesting that these hypo CHH hcDMRs in *mom1-3* and *pial1/2* mutants are RdDM sites (Supplementary Fig. 8a). Similarly, the hypo CHG hcDMRs in *mom1-3* and *pial/2* mutants are also mainly RdDM sites (Supplementary Fig. 8b). On the contrary, the majority of hypo CG hcDMRs in *mom1-3* and *pial1/2* mutants were barely enriched for Pol V ChIP-seq signal, were devoid of CHH and CHG methylation in Col-0, and were located in genes and in chromosome arms, suggesting that they are likely sites of gene body methylation (Supplementary Fig. 8c–e). In addition, only a very small proportion of hypo CG hcDMRs in *mom1-3* and in *pial1/2* double mutant overlapped (Supplementary Fig. 8f), suggesting that the majority of these hypo CG hcDMRs are unlikely due to the function of the MOM1 complex. It's likely that these hypo CG hcDMRs are accumulated random natural variations in CG methylation. Overall, these data suggests that the MOM1 complex helps maintain DNA methylation at some RdDM sites.

We next focused on CHH methylation to compare the effects of MOM1 complex and MORC mutants on RdDM sites. The hypo CHH hcDMRs in *mom1-3* and *pial1/2* notably overlapped with those of *morc6* and *morchex* at RdDM sites (520 DMRs in *morchex*)[39] (Fig. 5a, b, Supplementary Fig. 9, Supplementary Data 3). This is consistent with an earlier analysis that showed a strong overlap of *mom1* hypomethylated DMRs with those of the *morchex* mutant[7]. In addition, a small number of hypo CHH hcDMRs were detected in *mom2-2* (*n* = 23) and *aipp3-1* (*n* = 13), which also showed some overlap with those of the *morchex* mutant. Neither of the *phd1* mutant alleles tested showed any hypo CHH hcDMRs (Supplementary Data 3). To further explore the

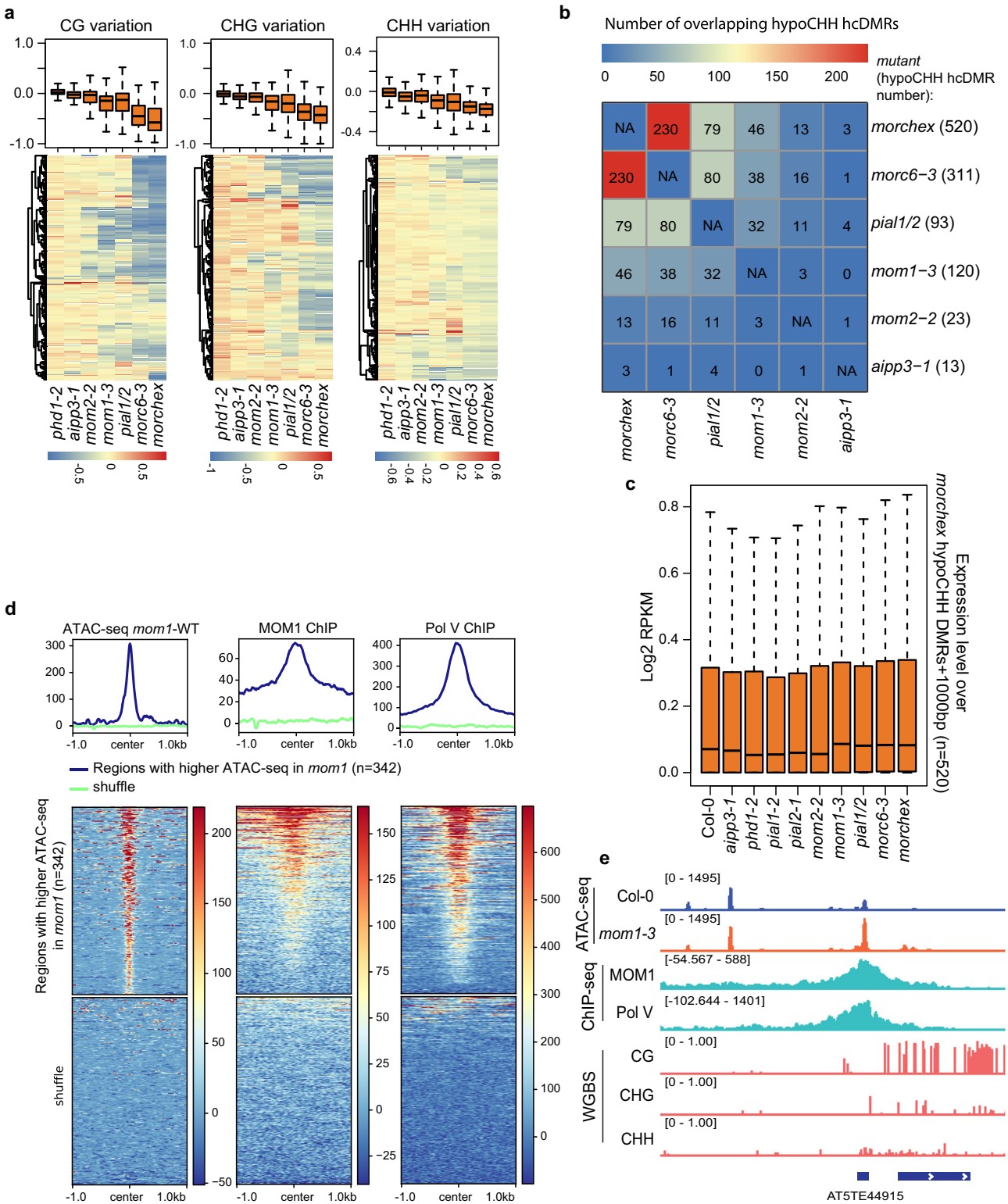

**Fig. 5 | The MOM1 complex influences DNA methylation and chromatin accessibility at some endogenous RdDM sites. a** Boxplots and heatmaps showing the variation of CG, CHG, and CHH DNA methylation in *phd1-2, aipp3-1, mom2-2, mom1-3, pial1/2, morc6-3* and *morchex* mutants vs Col-0 wild type over hypo CHH hcDMRs of the *morchex* mutant (*n* = 520). **b** the number and heatmap of overlapping of hypo CHH hcDMRs among *aipp3-1, mom2-2, mom1-3, pial1/2, morc6-3* and *morchex* mutants over *morchex* mutant hypo CHH hcDMRs (*n* = 520). **c** Boxplot representing the expression level (RNA-seq signal normalized by RPKM) of the genomic bins of 1 kb from hypo CHH hcDMRs (*n* = 520) of the *morchex* mutant in Col-0, *aipp3-1, phd1-2, pial1-2, pial2-1, mom2-2, mom1-3, pial1/2, morc6-3* and

*morchex* mutants. **d** Metaplots and heatmaps representing ATAC-seq signal (*mom1-3* minus Col-0), MOM1 ChIP-seq signal and Pol V ChIP-seq signal (subtracting control ChIP-seq signal) over regions with higher ATAC-seq signals in *mom1-3* (*n* = 342) and shuffled regions. **e** Screenshots of ATAC-seq signals of Col-0 and *mom1-3*, ChIP-seq signals of MOM1-Myc and Pol V (subtracting control signal) as well as CG, CHG, and CHH DNA methylation level by WGBS over a representative RdDM site. In box plots of **a** and **c**, center line represents the median; box limits represent the 25th and 75th percentiles; whiskers represent the minimum and the maximum. Source data are provided as a Source Data file.

functions of MOM1 complex components at these sites, we performed RNA-seq in Col-0, *morc6-3*, *morchex* and mutants of the MOM1 complex components. We observed that the expression level of the genomic regions within 1 kb of the 520 CHH hypo-DMR regions previously found in the *morchex* mutant were slightly upregulated in *mom1-3*, *pial1/2*, *morc6-3* and *morchex* mutants, but not in *phd1-2*, *aipp3-1*, *pial1-2*, *pial2-1*, or *mom2-2* mutants (Fig. 5c), showing that MOM1/PIAL1/PIAL2, along with MORCs, are required for the maintenance of CHH methylation and gene silencing at a small subset of RdDM sites, while AIPP3, PHD1, and MOM2 seem to play little role in this process.

Since MOM1-ZF is able to trigger early flowering in the *suvh2/9 fwa* background, it is possible that endogenously, the MOM1 complex is also able to recruit RdDM machinery in the absence of SUVH2/9. To test this hypothesis, we performed WGBS of the *suvh2/9* double mutant together with *mom1-3* and the Col-0 control. Consistent with a previous report[27], DNA methylation in the CHH context was lost in the *suvh2/9* double mutant over the majority of hypo CHH hcDMRs in the *nrpe1* mutant (Supplementary Fig. 10a), suggesting that the recruitment of the RdDM pathway by SUVH2/9 plays a predominant role at endogenous RdDM sites. Interestingly, 46 out of the 120 hypo CHH hcDMRs of the *mom1-3* mutant were not identified as hypo CHH hcDMRs in the *suvh2/9* double mutant. The CHH DNA methylation over these sites was largely preserved in the *suvh2/9* double mutant background (Supplementary Fig. 10b, c), suggesting that similar to the MOM1-ZF result, MOM1 is still able to trigger RdDM at these sites without SUVH2/9. At the same time, over the other 74 *mom1-3* hypo CHH hcDMRs (also identified as hypo CHH hcDMRs in *suvh2/9*), CHH DNA methylation was strongly decreased in the *mom1-3* mutant and in the *suvh2/9* double mutant (Supplementary Fig. 10d), suggesting that MOM1 and SUVH2/9 are both required for RdDM function at these loci.

We also performed ATAC-seq and detected 342 regions with increased ATAC-seq signal in the *mom1-3* mutant compared to Col-0 (Fig. 5d). As expected, these regions were enriched for MOM1 ChIP-seq signal (Fig. 5d). We also found that Pol V Chip-seq signal was highly enriched in these regions (Fig. 5d, e), suggesting that the MOM1 complex reduces chromatin accessibility at a subset of RdDM sites. Consistently, DNA methylation levels in CG, CHG and CHH contexts were decreased over the majority of these regions (Supplementary Fig. 11). Together, these results suggest that the MOM1 complex contributes to the endogenous function of the RdDM machinery, facilitating the maintenance of DNA methylation and a more closed chromatin state at some RdDM sites.

### The MOM1 complex has endogenous function divergent from the RdDM machinery

Previous studies have shown that the *mom1* mutants show derepression of pericentromeric heterochromatin regions, while the targets of the RdDM machinery tends to locate in euchromatic regions of the chromosome arms[5,35,48,49]. Consistent with these differences, we observed that ChIP-seq signals of MOM1, MORCs, and to a lesser extent PIAL2 were more highly enriched on transposable elements (TEs) located in pericentromeric regions as compared to TEs located in the chromosome arms – the opposite pattern to that of Pol V ChIP-seq[33] (Fig. 6a). From our RNA-seq, *mom1* and *pial1/2* mutants also showed transcriptional upregulation mainly in pericentromeric regions, while up-regulated TEs in the *nrpe1-11* mutant were located more broadly over the chromosomes including both pericentromeric regions and the euchromatic arms (Supplementary Fig. 12a). Consistent with previous reports[6], *morc6-3* and *morchex* mutants also displayed derepression of pericentromeric regions (Supplementary Fig. 12a). Upregulated differentially expressed TEs (DE-TEs) in the *morc6-3* and *morchex* mutants showed a prominent overlap with those of the *mom1-2*, *mom1-3*, and *pial1/2* mutants (Supplementary Fig. 12b). Meanwhile, many upregulated TEs in the *mom1-3* mutant are not derepressed or

only mildly derepressed in the *morc6-3* and *morchex* mutants, suggesting that the functions of the MOM1 complex and the MORC proteins do not fully overlap. The *phd1*, *aipp3*, and *mom2* mutants on the other hand showed little change in expression at these sites (Supplementary Fig. 12a, b), suggesting that these factors are less important for this silencing function.

We also discovered a set of MOM1 ChIP-seq peaks that did not overlap with DNA methylation (Supplementary Fig. 12c). We initially discovered these by performing unsupervised clustering of MOM1 ChIP-seq data with Pol V ChIP-seq data[33], and identified a group of MOM1 unique peaks not colocalizing with Pol V sites (Fig. 6b). We named the MOM1 and Pol V co-bound peaks as Cluster 1 peaks and the MOM1 unique peaks as Cluster 2 peaks (Fig. 6b). As expected, the Cluster 1 peaks were DNA methylated in all sequence contexts, while DNA methylation levels over Cluster 2 peaks were very low (Supplementary Fig. 12c). Other components of the MOM1 complex, such as the PIAL2, AIPP3 and to a lesser extent, PHD1 were also enriched at cluster 2 peaks (Fig. 6b). In addition, MORC4[40], MORC6 and MORC7[40] co-localized with MOM1 at both Cluster 1 and 2 peaks (Fig. 6b). Interestingly, we found that the Cluster 2 peaks were enriched for active histone marks H3K4me3 and H3Ac[50], as well as accessible chromatin indicated by ATAC-seq signal (Fig. 6c). This observation is consistent with a recent study reporting that MORC7 protein binds to active chromatin regions devoid of RdDM[40]. While H3K4me3 tends to peak after the Transcription Start Site (TSS), the MOM1 ChIP-seq signal tended to peak around the TSS of the genes near Cluster 2 peaks, similar to the ATAC-seq signal (Fig. 6d, e). The function of the MOM1 complex at these non-DNA methylated sites is currently unknown.

Overall, the ChIP-seq data suggests that while MOM1 and PIAL2 show strong localization to RdDM sites, they and the MORC proteins are more enriched in pericentromeric regions compared to the RdDM machinery. In addition, they are also present at unique active chromatin sites. The recruitment mechanism and the endogenous function of the MOM1 complex binding at the active chromatin sites need to be further investigated.

### Discussion

Due to the lack of major change in DNA methylation status in derepressed transgenes and endogenous TEs in the *mom1* mutant, MOM1 function has long been considered as independent of DNA methylation or downstream of DNA methylation. In our study, we observed a close link between the MOM1 complex and the RdDM machinery. By tethering the MOM1 complex with ZF in the *fwa* mutant, heritable DNA methylation was established at the *FWA* promoter, suggesting that the RdDM machinery was recruited as a result. Consistent with this, silencing and methylation of *FWA* were blocked in mutants of the DDR complex, as well as the *nrpe1* and *drm1/2* mutants, but not in the *suvh2/9* and *nrpd1* mutants. Thus, the recruitment of the de novo DNA methyltransferase DRM2 by the MOM1 complex requires the Pol V arm of the RdDM pathway. Previous MORC6-ZF tethering experiments resulted in similar results[43], i.e., the DDR complex and the downstream Pol V arm was required for silencing of *FWA*. In addition, we found that mutation of *MORC6* blocked *FWA* silencing mediated by ZF fusion to MOM1 complex components, suggesting that the MOM1 complex recruits the RdDM machinery via MORC6. This was also consistent with our observed physical interaction between PIAL2 of the MOM1 complex and MORC6. However, these observations do not exclude the possibility that physical interactions might also exist between MOM1 complex components and other components of the RdDM machinery.

We also found that MOM1 and PIAL1/2 are required for the efficiency of the establishment of methylation and silencing of *FWA* transgenes. Compared to RdDM mutants that completely block DNA methylation and silencing of *FWA* transgenes, the *mom1* and *pial1/2* mutants only showed a reduced efficiency of silencing, similar to what was observed in the *morchex* mutant. How the MOM1 complex

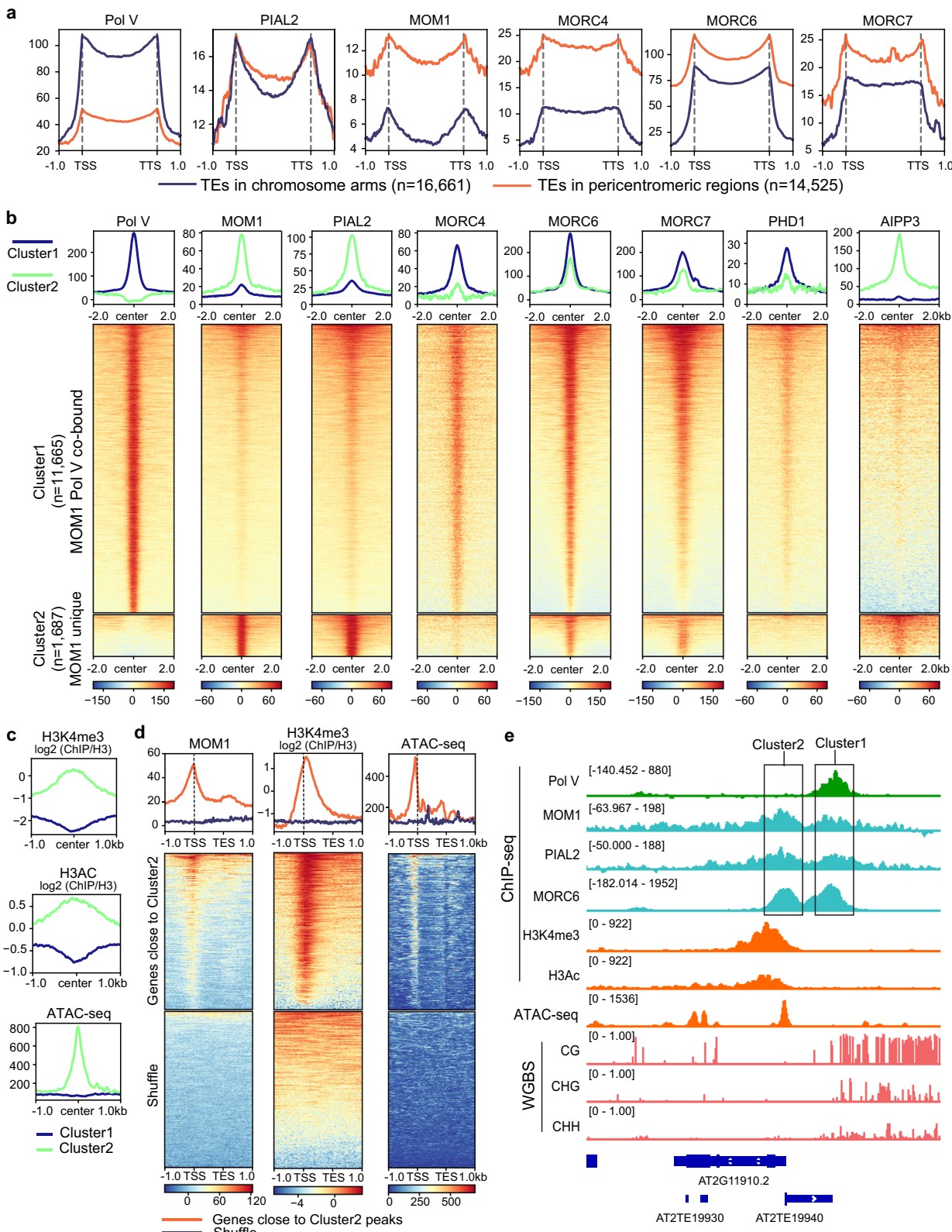

**Fig. 6 | MOM1 complex components and MORCs shows genomic distribution patterns distinct from that of the RdDM component Pol V. a** Metaplots of ChIP-seq signals of Pol V, PIAL2, MOM1, MORC4, MORC6, and MORC7 over TEs in euchromatic arms (*n* = 16,661) and TEs in pericentromeric regions (*n* = 14,525), with control ChIP-seq signals subtracted. **b** Metaplots and heatmaps of ChIP-seq signals of Pol V, MOM1, PIAL2, MORC4, MORC6, MORC7, PHD1, and AIPP3 over Cluster 1 and Cluster 2 ChIP-seq peaks of MOM1 and Pol V, with control ChIP-seq signals subtracted. **c**, Metaplots of ChIP-seq signals of H3K4me3 and H3Ac (normalized to H3),

as well as ATAC-seq signal of Col-0 over Cluster 1 and Cluster 2 peaks of MOM1 and Pol V. **d** Metaplots and heatmaps of MOM1 ChIP-seq signal (with control ChIP-seq signal subtracted), H3K4me3 ChIP-seq signal (normalized to H3) and ATAC-seq signal of Col-0 plants over genes close to Cluster 2 peaks and shuffled control regions. **e** Screenshots of Pol V, MOM1, PIAL2, MORC6 ChIP-seq signals with control ChIP-seq signals subtracted, H3K4me3 and H3Ac ChIP-seq signals, ATAC-seq signal of Col-0 plants, as well as CG, CHG, and CHH DNA methylation level by WGBS over a representative genomic region containing both Cluster 1 and Cluster 2 ChIP-seq peaks.

performs this function is unclear. The MOM1 complex might facilitate the initial loading of the RdDM machinery onto the *FWA* transgene, or it might allow for greater retention of the loaded RdDM machinery for more efficient DNA methylation and silencing, as has been suggested for the MORCs[40]. Consistent with the connections between MOM1 and RdDM revealed by ZF tethering results and *FWA* transgene silencing results, our ChIP-seq data showed that the MOM1 complex highly co-localized with RdDM sites in the genome. Our analysis of WGBS data also showed that MOM1 and PIAL1/2 were required to maintain CHH methylation at a small subset of RdDM sites, which notably overlap with CHH hypoDMR sites in the *morchex* mutants. A previous study also reported a similar observation with WGBS data from a different *mom1* mutant allele (*mom1-2*)[7]. Thus, aside from the previous findings that that transgene and TE silencing are released in the *mom1* mutant background without major DNA methylation changes[3,5,6], the MOM1 complex[7], together with the MORC proteins, are also required for the maintenance of DNA methylation at a small subset of RdDM sites. It seems likely that this would be mechanistically related to the role of both MOM1 and MORCs in the establishment of *FWA* transgene silencing, and it is intriguing to speculate that this might reflect an ancient role of these proteins in the initial establishment of methylation and silencing of novel invading transposable elements. We also found that MORC6 ChIP-seq signal at RdDM sites was strongly decreased in the *mom1* and *pial1/2* mutants, suggesting that MORC6 is loaded onto endogenous RdDM sites by the MOM1 complex, together with other mechanisms[29,37,51]. Based on these observations, we propose a model (Fig. 7) for how the MOM1 complex influences the RdDM machinery as follows. The MOM1 complex is first loaded onto RdDM target sites through an unknown mechanism to facilitate the binding of the MORC6 protein. MORC6 would then enhance the recruitment the Pol V arm of the RdDM machinery to methylate target DNA, by topologically entrapping the DNA as well as directly interacting with RdDM components, thus serving as a tethering factor[29,37,51,52].

From previous studies[4,5] and data from this study, it seems clear that the MOM1 complex has at least two functions in epigenome

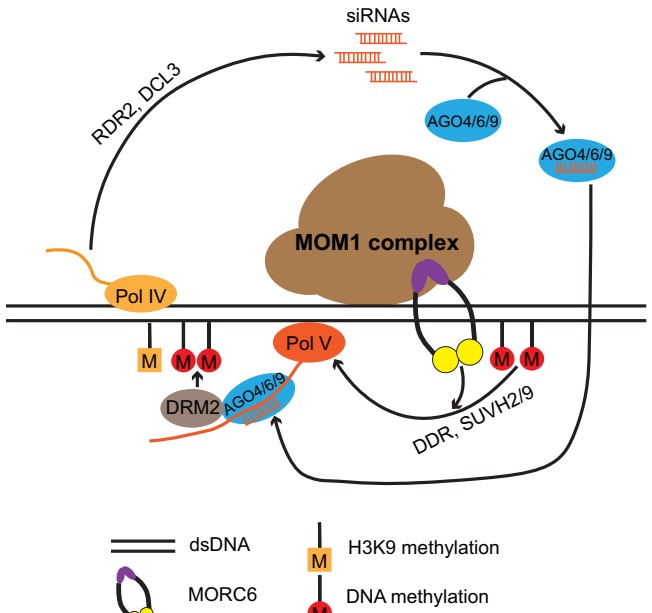

**Fig. 7 | Working model of MOM1 complex.** The MOM1 complex is first loaded onto RdDM target sites through an unknown mechanism to facilitate the binding of the MORC6 protein. MORC6 would then enhance the recruitment the Pol V arm of the RdDM machinery to methylate target DNA, by topologically entrapping the DNA as well as directly interacting with RdDM components, thus serving as a tethering factor[29,37,51,52].

regulation, a role in the establishment and maintenance of RdDM, and a role in the maintenance of silencing of TEs in pericentromeric regions. Furthermore, it appears that these two functions are mechanistically distinct. For example, comparison of DE-TEs and DE-genes in the *nrpe1* and *mom1* mutants in previous studies[5,35] indicates that the majority of their endogenous targets do not overlap. Thus, for instance, the loss of RdDM function in the *nrpe1* mutant does not impair the silencing function of MOM1 at the majority of its TE targets. It's possible that the localization of MOM1 at RdDM is only needed for the silencing of the relatively small number of shared TE targets between MOM1 and RdDM. In future studies, it will be interesting to investigate the relationship between the two functions of the MOM1 complex, and identify the MOM1 complex component(s) or protein domain(s) that might be required for only one of the functions.

In addition to the localization at RdDM sites, we identified a unique set of MOM1 peaks which are enriched with active chromatin marks. This is reminiscent of an earlier study reporting that MOM1 regulates transcription in intermediate heterochromatin, which is associated with both active and repressive histone marks[48]. Interestingly, the MOM1 complex and MORCs seem to behave similarly in binding active chromatin, as MORC7 was also reported to bind active chromatin devoid of RdDM[53], and MORCs are colocalized at these MOM1 unique peaks. The mechanism of recruiting the MOM1 complex to these unique peaks and the function of MOM1 at these active chromatin sites is unknown.

In summary, our results uncover the function for the MOM1 complex in the efficiency of both the establishment and maintenance of RNA-directed DNA methylation and gene silencing, and point to a potential function at some unmethylated euchromatic regions, suggesting that MOM1 plays multifaceted roles in epigenome regulation.

## Methods

### Growth condition, molecular cloning and plant materials

*Arabidopsis thaliana* plants in this study were Col-0 ecotype and were grown under 16 h light: 8 h dark condition. The T-DNA insertion lines used in this study are: aipp3-1 (GABI_058D11), aipp3-2 (SAIL_1246_E10), mom1-2 (SAIL_610_G01), mom1-3 (SALK_141293), mom1-7 (GABI_815H11), mom2-1 (WiscDsLox364H07), mom2-2 (SAIL_548_H02), pial1-2 (CS358389), pial2-1 (SALK_043892), morc6-3 (GABI_599B06), aipp2-1 (SALK_057771), suvh2 (SALK_079574), suvh9 (SALK_048033), nrpe1-11 (SALK_029919) and *morchex*[39] consisting of morc1-2 (SAIL_893_B06), morc2-1 (SALK_072774C), morc4-1 (GK-249F08), morc5-1 (SALK_049050C), morc6-3 (GABI_599B06), and morc7-1 (SALK_051729). In addition to the T-DNA insertion line, three *phd1* mutant alleles were generated using a YAO promoter driven CRISPR/Cas9 system[54]. *phd1-2* contained a single nucleotide T insertion and *phd1-3* contained a 13-nucleotide deletion and an 18-nucleotide duplication in the 2nd exon of PHD1 gene, both of which led to early termination of the protein at amino acid 53 located within the PHD domain. *phd1-4* contained a single nucleotide T insertion in the 3rd exon of the PHD1 gene, leading to early termination of the PHD1 protein at amino acid 88. The *fwa* background RdDM mutants, including nrpd1-4 (SALK_083051), suvh2 (SALK_079574) suvh9 (SALK_048033), morc6-3 (GABI_599B06), rdm1-4 (EMS)[55], drd1-6 (EMS)[56], dms3-4 (SALK_125019C), nrpe1-1 (EMS), and drm1-2 (SALK_031705) drm2-2 (SALK_150863) were described by Gallego-Bartolomé et al.[43]. The other *fwa* background mutants in MOM1 complex were phd1-2, aipp3-1 (GABI_058D11), mom1-3 (SALK_141293), mom2-1 (WiscDsLox364H07), and pial1 (CS358389) pial2 (SALK_043892), which were generated by crossing *fwa-4* to corresponding mutants. F2 offspring plants with late flowering phenotype were genotyped for homozygous T-DNA mutant alleles, and propagated to F3 generation. Then, F3 populations were screened for non-segregating homogenous late flowering phenotype. For IP-MS comparisons of MOM1-FLAG in *mom1-7* mutant background, to that in the backgrounds of *aipp3-1*, *mom2-2*, as well as *aipp3/mom2-2*

double mutants, MOM1-FLAG transgenic lines were constructed by recombineering 2xYpet-3xFLAG encoding DNA sequence in frame with the C terminus of MOM1 gene, in a transformation-competent artificial chromosome clone (JAtY68M20 (68082 bp)) using a bacterial recombineering approach[57] and transformed into *mom1-7* mutants. Then this MOM1-FLAG transgenic line was crossed into *aipp3-1*, *mom2-2*, as well as *aipp3/mom2-2* double mutant backgrounds. For transgenic plants of FLAG epitope tagged, Myc epitope tagged and ZF tagged proteins used in all other IP-MS, ChIP-seq and ZF tethering experiments, genomic DNA fragments including the promoter region were amplified and cloned into entry vectors (pENTR-D or PCR8 from Invitrogen) and cloned into destination vectors with C-terminal 3xFLAG (pEG302_GW_3xFLAG), Myc (pEG302_GW_9x Myc) and ZF108 (pEG302_GW_3xFLAG_ZF108) by LR clonase II (Invitrogen). Primers used in this study were listed in Supplementary Data 4. *Agrobacterium* mediated floral dipping (strain Agl0) were used to generate transgenic plants in corresponding loss-of-function mutant backgrounds or specific mutant backgrounds as indicated.

## IP-MS and cross-linking IP-MS

50 mL of liquid nitrogen flash-frozen unopened flower buds from FLAG epitope tagged transgenic plants were used for each IP-MS experiment and flower buds of Col-0 plants were used as control. Flower tissue was ground to fine powder in liquid nitrogen with Retsch homogenizer. For Native IP-MS, tissue powder was resuspended in 25 mL IP buffer (50 mM Tris-HCl pH 8.0, 150 mM NaCl, 5 mM EDTA, 10% glycerol, 0.1% Tergitol, 0.5 mM DTT, 1 mg/mL Pepstatin A, 1 mM PMSF, 50 μM MG132 and cOmplete EDTA-free Protease Inhibitor Cocktail (Roche)) and further homogenized with dounce homogenizer. The lysates were filtered with Miracloth and centrifuged at 20,000 g for 10 min at 4 °C. The supernatant was incubated with 250 μL anti-FLAG M2 magnetic beads (Sigma) at 4 °C for 2 h with constant rotation. The magnetic beads were washed with IP buffer and eluted with 250 μg/mL 3xFLAG peptides. Eluted proteins were used for Trichloroacetic acid (TCA) precipitation and mass spectrometric analysis.

For Crosslinking IP-MS, flower tissue powder was resuspended in 40 mL nuclei extraction buffer[40] with 1.5 mM EGS (Ethylene Glyco-bis (succinimidylsuccinate)) and rotated at room temperature for 10 min. Then the lysate was supplemented with formaldehyde at 1% final concentration and rotated at room temperature for another 10 min followed by adding glycine to stop crosslinking. The crosslinked lysate was filtered through Miracloth and centrifuged for 20 min at 2880 g. The pellet (which contains the nuclei) was resuspended in 3 mL of extraction buffer 2 (0.25 M sucrose, 10 mM Tris-HCl pH 8.0, 10 mM MgCl₂, 1% Triton X-100, 5 mM 2-Mercaptoethanol, 0.1 mM PMSF, 5 mM Benzamidine and cOmplete EDTA-free Protease Inhibitor Cocktail (Roche)), then centrifuged at 12,000 g for 10 min at 4 °C. Then, the pellet was carefully resuspended in 1.2 mL nuclear lysis buffer (50 mM Tris-HCl pH 8.0, 10 mM EDTA, 1% SDS, 0.1 mM PMSF, 5 mM Benzamidine and cOmplete EDTA-free Protease Inhibitor Cocktail (Roche)) and incubated on ice for 10 min. After that, 5.1 mL dilution buffer (1.1% Triton x-100, 1.2 mM EDTA, 16.7 mM Tris-HCl pH 8.0, 167 mM NaCl, 1 mM PMSF, 5 mM Benzamidine and cOmplete EDTA-free Protease Inhibitor Cocktail (Roche)) was added and mixed by pipetting. Resuspended nuclei were split into 3 × 2.1 mL aliquots for sonication of 22 min (30 s on/30 s off) with Bioruptor Plus (Diagenode). Sheared lysate from the same sample was combined and centrifuged at 12,000 g for 10 min at 4 °C. Another 6 mL of dilution buffer and 250 μL anti-FLAG M2 magnetic beads (Sigma) were added to the supernatant and the sample was incubated at 4 °C for 2 h with constant rotation. Then, the magnetic beads were washed and eluted with 250 μg/mL 2xFLAG peptides. Eluted protein was used for Trichloroacetic acid (TCA) precipitation and mass spectrometric analysis.

MS/MS database searching was performed using MaxQuant (1.6.10.43) against newest *Arabidopsis thaliana* proteome database

[http://www.uniprot.org]. Analysis of raw data was obtained from the LC−MS runs using MaxQuant with the integrated Andromeda peptide search engine using default setting with enabled LFQ normalization. Data sets were filtered at a 1% FDR at both the PSM and protein levels. The MaxQuant peptide intensity and MS/MS counts were used for all peptide quantitation. For Fig. 1c, fold of change of MS/MS counts and *P* value of MOM1-FLAG lines crosslinking IP-MS compared to crosslinking IP-MS of Col-0 control were calculated by LIMMA[58] (v3.52.4).

## Chromatin immunoprecipitation sequencing

For chromatin immunoprecipitation sequencing (ChIP-seq), 15 mL of unopened flower buds were collected for each ChIP and flash-frozen in liquid nitrogen. The flower tissue was ground to fine powder with Retsch homogenizer in liquid nitrogen and resuspended in nuclei extraction buffer (50 mM HEPES pH 8.0, 1 M sucrose, 5 mM KCl, 5 mM MgCl₂, 0.6% Triton X-100, 0.4 mM PMSF, 5 mM benzamidine, cOmplete EDTA-free Protease Inhibitor Cocktail (Roche), 50 μM MG132). For transgenic lines of MOM1-Myc in *mom1-7* and PIAL2-Myc in *pial2-1*, EGS was first added to resuspended lysate to 1.5 mM and the tissue lysate was incubated at room temperature for 10 min with rotation. Then the lysate was supplemented with formaldehyde at 1% and rotated at room temperature for another 10 min followed by adding glycine to stop crosslinking. For ChIP of all other proteins, crosslinking was performed by directly supplementing formaldehyde to 1% without adding EGS, then rotated at room temperature for 10 min followed by adding glycine to stop crosslinking. The crosslinked nuclei were isolated, lysed with Nuclei Lysis Buffer and diluted with ChIP Dilution Buffer[40]. Then the lysate was sonicated for 22 min (30 s on/30 s off) with Bioruptor Plus (Diagenode). After centrifugation, antibody for FLAG epitope (M2 monoclonal antibody, Sigma F1804, 10 μL per ChIP added at a final dilution of 1:400) or for Myc epitope (Cell Signaling, 71D10, 20 μL per ChIP added at a final dilution of 1:200) were added to the supernatant and incubated at 4 °C overnight with rotation. Then, Protein A and Protein G Dynabeads (Invitrogen) were added and incubated at 4 °C for 2 hours with rotation. After that, the beads were washed and eluted, and the eluted chromatin was reverse-crosslinked by adding 20 μL 5 M NaCl and incubated at 65 °C overnight followed by treatment of Proteinase K (Invitrogen) for 4 hours at 45 °C. DNA was purified and precipitated with 3 M Sodium Acetate, GlycoBlue (Invitrogen) and ethanol at −20 °C overnight. After centrifugation, the precipitated DNA was washed with ice cold 70% ethanol, air dried and dissolved in 120 μL of H₂O. ChIP-seq libraries were prepared with Ovation Ultra Low System V2 kit (NuGEN), and sequenced on Illumina NovaSeq 6000 or HiSeq 4000 instruments.

For ChIP-seq analysis, raw reads were trimmed using trim_galore (https://www.bioinformatics.babraham.ac.uk/projects/trim_galore/) and aligned to the TAIR10 reference genome (https://www.arabidopsis.org/download/index-auto.jsp%3Fdir%3D%252Fdownload_files%252FGenes%252FTAIR10_genome_release) with bowtie2 (v2.4.2)[59] allowing zero mismatch and reporting one valid alignment for each read. The Samtools (v1.15)[60] were used to convert sam files to bam files, sort bam files and remove duplicate reads. Track files in bigWig format were generated using bamCoverage of deeptools (v3.5.1)[61] with RPKM normalization. Peaks were called with MACS2 (v2.1.2)[62] and peaks frequently identified in previous ChIP-seq of Col-0 plant with M2 antibody for FLAG epitope were removed from analysis.

For unsupervised clustering of Pol V and MOM1 peaks (Fig. 6b), RPKM of Pol V[33], MOM1 and corresponding control ChIP-seqs over merged peaks of Pol V and MOM1 were calculated with custom scripts. Then, log₂(PolV RPKM/control RPKM) and log₂(MOM1 RPKM /control RPKM) were calculated and used for unsupervised clustering with the ConsensusClusterPlus R package (v1.60.0)[63]. For analysis of ChIP signal over TEs located in euchromatic arms versus TEs located in pericentromeric regions (Fig. 6a), the pericentromeric regions were defined by Bourguet et al. [64].

## RNA sequencing

For RNA-seq experiments, twelve-day old seedlings grown on half MS medium (Murashige and Skoog Basal Medium) were collected and flash-frozen in liquid nitrogen. RNA was extracted with Direct-zol RNA MiniPrep kit (Zymo Research) and 1 µg of total RNA was used to prepare RNA-seq libraries with TruSeq Stranded mRNA kit (Illumina), and the libraries were sequenced on Illumina NovaSeq 6000 instruments.

The raw reads of RNA-seq were aligned to the TAIR10 reference genome with bowtie2. Rsem-calculate-expression (v1.3.1) from RSEM[65] with default settings was used to calculate expression levels. DEGs and DE-TEs were calculated with "run_DE_analysis.pl" from Trinity version 2.8.5[66] and $\log_2 FC \geq 1$ and FDR < 0.05 were used as the cut off. RNA-seq track files in bigWig format were generated using bamCoverage of deeptools (v3.1.3) with RPKM normalization.

## Whole genome bisulfite sequencing

Rosette leaves of about one-month-old *Arabidopsis* Col-0 wild type, *phd1-2*, *phd1-3*, *mom2-2*, *aipp3-1*, *fwa* plants and ZF transgenic lines (MOM1-ZF, MOM2-ZF, PIAL1-ZF, PIAL2-ZF and PHD1-ZF) T2 plants with early flowering phenotype were collected for DNA extraction using DNeasy Plant Mini Kit (QIAGEN). 500 ng DNA was sheared with Covaris S2 (Covaris) into around 200 bp at 4 °C. The DNA fragments were used to perform end repair reaction using the Kapa Hyper Prep kit (Roche), and together with Illumina TruSeq DNA sgl Index Set A/B (Illumina) to perform adapter ligation. The ligation products were purified with AMPure beads (Beckman Coulter), and then converted with EpiTect Bisulfite kit (QIAGEN). The converted ligation products were used as templates, together with the primers from the Kapa Hyper Prep kit (Roche) and MyTaq Master mix (Bioline) to perform PCR. The PCR products were purified with AMPure beads (Beckman Coulter) and sequenced by Illumina NovaSeq 6000 instrument.

The WGBS data analysis was performed by aligning the raw reads to both strands of the TAIR10 reference genome using BSMAP (v.2.74)[67], allowing up to 2 mismatches and 1 best hit. Reads with more than 3 consecutive methylated CHH sites were removed, and the methylation level was calculated with the ratio of $C/(C + T)$. For Fig. 2d, the methylation levels at 1 kb flanking regions of ZF off target sites[43] in MOM1-ZF, MOM2-ZF, PIAL1-ZF, PIAL2-ZF and PHD1-ZF were subtracted by the methylation level of *fwa* and plotted with R package pheatmap (v1.0.12).

The hcDMRs ($p < 0.01$, > 33 supported controls) of *mom1-3*, *pial1 pial2*, *aipp3-1*, *phd1-2*, *mom1-2*, *mom2-1*, *morc6*, and *morchex* mutants were called[7]. For Supplementary Fig. 9, the negative natural log of *P*-value for hypo CHH hcDMRs overlaps was calculated by HOMER[68] (v4.11.1) mergePeaks using hypergeometric distribution. Col-0 DNA methylation tracks used in screenshots were from dataset GSM3553007[43].

## BS-PCR-seq

Rosette leaves of about one-month-old plants were collected and subject to DNA extraction with CTAB method followed by bisulfite DNA conversion using the EpiTect Bisulfite kit (QIAGEN) kit. Three regions of the *FWA* gene were amplified from the converted DNA with Pfu Turbo Cx (Agilent): Region 1 (chr4: 13038143-13038272), Region 2 (chr4: 13038356- 13038499) and Region3 (chr4: 13038568-13038695). Primers used are listed in Supplementary Data 4. Libraries were prepared with the purified PCR product by the Kapa DNA Hyper Kit (Roche) together with TruSeq DNA UD indexes for Illumina (Illumina) and were sequenced on Illumina iSeq 100 or HiSeq 4000 instruments.

BS-PCR-seq data was analyzed by aligning the raw reads to both strands of the TAIR10 reference genome with BSMAP (v.2.90)[67] allowing up to 2 mismatches and 1 best hit. After quality filtering, the methylation level of cytosines was calculated as the ratio of $C/(C + T)$, and customized R scripts were used to plot methylation data over the *FWA* region 1-3.

## ATAC-seq

Fresh unopened flower buds of about one-month-old Col-0 and *mom1-3* mutant plants were collected for nuclei extraction and ATAC-seq, with two replicates for each genotype. We collected nuclei from unopened flower buds[33], which were used for ATAC-seq[69]. Unopened flower buds were collected for extraction of nuclei as follows. About 5 grams of unopened flower buds was collected and immediately transferred into ice-cold grinding buffer (300 mM sucrose, 20 mM Tris pH 8, 5 mM MgCl₂, 5 mM KCl, 0.2% Triton X-100, 5 mM β-mercaptoethanol, and 35% glycerol). The samples were ground with Omni International General Laboratory Homogenizer on ice and then filtered through a two-layer Miracloth and a 40-µm nylon mesh Cell Strainer (Fisher). Samples were spin filtered for 10 min at 3,000 $g$, the supernatant was discarded, and the pellet was resuspended with 25 mL of grinding buffer using a Dounce homogenizer. The wash step was performed twice in total, and nuclei were resuspended in 0.5 mL of freezing buffer (50 mM Tris pH 8, 5 mM MgCl₂, 20% glycerol, and 5 mM β-mercaptoethanol). Nuclei were subjected to a transposition reaction with Tn5 (Illumina). For the transposition reaction, 25 µL of 2× DMF (66 mM Tris-acetate pH 7.8, 132 mM K-Acetate, 20 mM Mg-Acetate, and 32% DMF) was mixed with 2.5 µL Tn5 and 22.5 µL nuclei suspension at 37 °C for 30 min. Transposed DNA fragments were purified with ChIP DNA Clean & Concentrator Kit (Zymo). Libraries were prepared with Phusion High-Fidelity DNA Polymerase (NEB) in a system containing 12.5 µL 2x Phusion, 1.25 µL 10 mM Ad1 primer, 1.25 µL 10 mM Ad2 primer, 4 µL ddH2O, and 6 µL purified transposed DNA fragments. The ATAC-seq libraries were sequenced on HiSeq 4000 platform (Illumina).

For ATAC-seq data analysis, raw reads were adaptor-trimmed with trim_galore and mapped to the TAIR10 reference genome with Bowtie2[59] (-X 2000 -m 1). After removing duplicate reads and reads mapped to chloroplast and mitochondrial, ATAC-Seq open chromatin peaks of each replicate were called using MACS2 with parameters "-p 0.01–nomodel–shift −100–extsize 200". Consensus peaks between replicates were identified with bedtools (version 2.26.0) intersect and differential accessible peaks were called with the R packge edgeR[70] (version 3.30.0). Merged bigwig file of the two replicates were used for heatmap and metaplot.

## RT-qPCR

Rossette leaves of about one-month-old plants were collected for RNA extraction with Zymo Direct-Zol RNA miniprep Kit (Zymo Research). A total of 1 µg of RNA were used for cDNA synthesis with iScript cDNA Synthesis Kit (Bio-Rad). qPCR was performed with iQ SYBR Green Supermix (Bio-Rad) and primers for qPCR were listed in Supplementary Data 4.

## McrBC assay

Genomic DNA extracted with the CTAB method were treated with RNase A (Qiagen) and diluted to about 100 ng/µL. 10 µL of diluted DNA were used for McrBC digestion (NEB, 4 h at 37 °C) or mock digestion (the same volume of H₂O instead of McrBC enzyme was added with all other components the same in the reaction, was also kept for 4 h at 37 °C). Relative undigested *FWA* promoter quantity (McrBC treated/H₂O treated) was determined with qPCR and primers used were listed in Supplementary Data 4.

## Flowering time measurement

Total true leaf numbers (sum of rosette leaf number and cauline leaf number) after bolting of the plants were used as measurement of flowering time. Plants with less than 20 true leaf number were considered as early flowering. Detailed flowering times as raw leaf count for each plant are listed in Source Data of Figs. 2a, 3a, 4a, c, Supplementary Fig. 2a, Supplementary Fig. 2c, Supplementary Fig. 3c, Supplementary Fig. 4a–d, and Supplementary Fig. 7b. The numbers of

independent plants (*n*) scored for each population and detailed statistics of flowering time comparison between different populations are listed in Supplementary Data 5.

### Yeast two-hybrid (Y2H)

The cDNA sequences of PIAL1, PIAL2, MOM2, MORC6, and MOM1 CMM2 domain (aa1660-aa1860)[5] were first cloned into gateway entry vectors followed by LR reaction with pGBKT7-GW (Addgene 61703) and pGADT7-GW (Addgene 61702) destination vectors. Pairs of plasmid DNA for the desired protein interaction to be tested were co-transformed into the yeast strain AH109. Combinations of the empty pGBKT7-GW or pGADT7-GW vectors and the plasmids of desired proteins were used for transformation of yeast cells to test for self-activation. Transformed yeast cells were plated on synthetic dropout medium without Trp and Leu (SD-TL) and incubated for 2–3 days to allow for the growth of positive colonies carrying both plasmids. Three yeast colonies of each tested protein interaction pairs were picked and mixed in 150 μL 1×TE solution, and 3 μL of the 1×TE solution with the yeast cells were blotted on synthetic dropout medium without Trp, Leu, and His (SD-TLH) and with 5 mM 3-amino-1,2,4-triazole (3AT) to inhibit background growth. Growth of yeast on SD-TLH with 5 mM 3AT medium after 2–3 days of incubation indicates the interaction between the GAL4-AD fusion protein and the GAL4-BD fusion protein.

### Co-immunoprecipitation

A total of 2 grams of 2-week-old seedling tissue were collected from MORC6-FLAG X PIAL2-Myc F1 generation and PIAL2-Myc transgenic plants and ground into fine powder in liquid nitrogen. The tissue powder was resuspended with 10 mL IP buffer, and incubated for 20 min at 4 °C. Then the lysate was centrifuged and filtered with Miracloth twice. A total of 30 μL of anti-FLAG M2 Affinity Gel (Millipore) was added to the supernatant and incubated for 2 h at 4 °C. Then, the anti-FLAG beads were washed with IP buffer for 5 times, and eluted with 40 μL elution buffer (IP buffer with 100 μg/mL 3xFLAG peptide). The eluted protein was used for western blot. Anti-Myc/c-Myc antibody (9E10) HRP (Santa cruz Biotechnology sc-40 HRP, 1:3000 dilution) and monoclonal ANTI-FLAG M2 HRP (Sigma-Aldrich A8592, 1:7500 dilution) were used for western blot.

### Reporting summary

Further information on research design is available in the Nature Portfolio Reporting Summary linked to this article.

## Data availability

The high-throughput sequencing data generated in this study have been deposited in the National Center for Biotechnology information Gene Expression Omnibus database under accession code GSE221679. The mass spectrometry proteomics data generated in this study have been deposited in the ProteomeXchange Consortium via the PRIDE[71] partner repository under accession code PXD039991. The TAIR10 reference genome used in this study are available at The Arabidopsis Information Resource website [https://www.arabidopsis.org/download/index-auto.jsp%3Fdir%3D%252Fdownload_files%252FGenes%252FTAIR10_genome_release]. The Col-0 DNA methylation data used in this study for screenshots in Figs. 1b, 3c, 5e, 6e and Supplementary Fig. 6b are available in the National Center for Biotechnology information Gene Expression Omnibus database under accession code GSE124746. The DNA methylation data of the *nrpe1* mutant and corresponding Col-0 control plants used in this study for Supplementary Fig. 8a–c and Supplementary Fig. 10b–d are available in the National Center for Biotechnology information Gene Expression Omnibus database under accession code GSE39901. The Col-0 DNA methylation data used in this study for Supplementary Fig. 12c are available in the National Center for Biotechnology information Gene Expression Omnibus database under accession code GSE54677. The Col-0 DNA

methylation data used in this study for Supplementary Fig. 3a are available in the National Center for Biotechnology information Gene Expression Omnibus database under accession code GSE80302 . Source data are provided as a Source Data file. Source data are provided with this paper.

## Code availability

The customized code used in this study have been deposited in the GitHub repository [https://github.com/Zhenhuiz/MOM1_NC_2023].

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

## Acknowledgements

We thank Suhua Feng and Mahnaz Akhavan for support with high-throughput sequencing at the UCLA Broad Stem Cell Research Center BioSequencing Core Facility. This work was supported by NIH R35 GM130272 to S.E.J. S.E.J. is an Investigator of the Howard Hughes Medical Institute.

## Author contributions

Z.L., M.W., Z.Z., and S.E.J designed the research, interpreted the data, and wrote the manuscript; Z.L., M.W., and Z.Z. performed experiments and performed bioinformatic data analysis; Y.J.A. and J.W. performed IP-MS and interpreted the data. S.B. and J.A.L. contributed to gathering mutant materials, construction of transgenic lines, performing initial ZF108 tethering assays and discussions. J.G.B. contributed to PHD1- ZF108 materials. S.F. performed BS-PCR-seq and high throughput sequencing; X.W. provided technical support.

## Competing interests

The authors declare no competing interests.
