## [Peer Review File · Nature Communications]

The MOM1 complex recruits the RdDM machinery via MORC6 to establish de novo DNA methylation.Reviewers' Comments:

Reviewer #1:

Remarks to the Author:

In this manuscript entitled "The MOM1 complex recruits the RdDM machinery via MORC6 to establish de novo DNA methylation", the authors explored the function of MOM1 complex in establishing and maintaining DNA methylation in Arabidopsis. To explore the function of MOM1, the authors performed ChIP-seq of MOM1 and MOM1 complex components. Unexpectedly, these proteins were all colocalized with Pol V at RdDM sites, suggesting the MOM1 complex may also function in DNA methylation and gene silencing. Using the well-established *fwa* silencing system, the authors demonstrated that tethering of MOM1 complex components to the FWA promoter in the *fwa* mutant by ZF fusion led to the establishment of DNA methylation and silencing of the FWA gene that depends on the Pol V arm of the RdDM pathway and MORC6. The MOM1 complex is also required to maintain a small number of endogenous RdDM target loci. The authors also demonstrated that PIAL2 could interact with MORC6. Based on the genetic results, they proposed that the MOM1 complex recruits the RdDM machinery via MOR6 to establish de novo DNA methylation. Furthermore, the authors identified a group of MOM1-specific targets at active chromatin regions, although the function of MOM1 in these regions is still unclear. In summary, this study reported a new function of MOM1 in DNA methylation and helped explore the role of MOM1 in gene silencing. Overall, this manuscript was well-organized and easy to follow.

Comments:

1. Please perform statistical analysis for flowering time in Fig.2a, Fig. 3a, Fig. 4a and 4c et al.
2. To better support the conclusion, the authors should perform MORC6 ChIP-seq in the *mom1* background.
3. In Fig. 5d, the authors performed ATAC-seq in *mom1* and WT and identified 342 regions with increased ATAC-seq signal in the *mom1-3* mutant background. Do these 342 regions overlap with MOM1-binding sites? How many sites are overlapped between 342 regions with CHH-hypo sites?
4. In Fig.2d, please include the Col-0 and *fwa* controls.
5. A working model should be proposed to explain the function of MOM1 easily.

Reviewer #2:

Remarks to the Author:

Mechanisms that MOM1 affects gene silencing remain enigmatic since the discovery of this gene. Here the authors show that MOM1 functions upstream of MORC and affects silencing and DNA methylation in RdDM targets. That is a new aspect of gene silencing mechanisms and very nice extension of the results of the Jacobsen group on MORC and RdDM. The effects of factors in this new pathway have been extensively studied in this paper. The results are novel and look convincing overall. I have only minor suggestions.

- 1) It seems that the MOM1 complex has at least two functions, RdDM in euchromatic regions and transcriptional silencing of heterochromatic regions. I wonder if these two functions are connected or not. For example, as the authors mentioned briefly in Discussion for the FWA locus, is it possible that the *mom1* mutation compromises RdDM genome-wide via transcriptional derepression?
- 2) For the same reason, I wonder if the authors can detect differential effects of modified proteins of MOM1 complex for these two pathways. As the authors regard MOM1 has "multifaceted roles" (the last sentence of the Abstract) it would be nice if they can detect region(s) necessary for one of these roles but not for the other.
- 3) It seems that the MOM1 pathway and SUVH2/9 pathway may induce RdDM in parallel. I wonder how epigenome dynamics changes in the *mom1 suvh2/9* triple mutant.

Reviewer #3:

Remarks to the Author:

The manuscript entitled "The MOM1 complex recruits the RdDM machinery via MORC6 to establish de novo DNA methylation" by Li et al. reports a link between the MOM1 complex and RdDM machinery, as well as MORC proteins in Arabidopsis. The authors performed Mass-spec analysis to identify proteins interacting with MOM1 and identified core proteins, including MOM1/2, PIAL1/2, PHD1, and AIPP3. The authors further examined the role of the MOM1 complex in de novo DNA methylation and epistasis with RdDM and MORC genes using the FWA silencing assay, demonstrating the involvement of the MOM1 complex in RdDM-dependent silencing of the FWA locus. In addition, they showed MOM1 localizations independent of RdDM target loci.

Overall, the experiments performed in this work were well designed, and conclusions are very clear and supported by the rigorous results, including those with MASS-spec and transgenic assays. The findings, including the role of MOM1, which was long thought as independent of DNA methylation, in RdDM and MORC pathways are novel and interesting, and would have a broad impact in the field. I would recommend this manuscript for publication in Nature Communications, provided the authors address the following points.

-Figure 3a: It looks like the introduction of MOM1-ZF to *drm1/2 fwa* still induced early flowering in some transgenic T1 plants. Is there any possibility of RdDM independent silencing mechanisms of FWA gene such as H3K9me or PTGS in this system?

-Line 326-333: Even though the description of overlaps of CHH DMRs between *morchex* and *mom1*, *pail1/2*, *aipp3* etc., no data was shown for the exact number of the overlapping DMRs. Sup. Table 3 shows only the total numbers and positions of DMRs. The number of overlapping DMRs or Venn diagram can be provided.

-Line 366-367; "We also discovered a set of MOM1 ChIP peaks that did not overlap with DNA methylation". I think the data of DNA methylation levels associated with the MOM1 ChIP peaks were not shown in this manuscript.

-Figure 5a: Although the manuscript mainly focused on CHH methylation, *mom1* and *pail1/2* showed variations in CGm and CHGm in quite some DMRs. Are these DMRs associated with TEs in pericentromeric regions?

-Figure 6, Line 437-439: While it is clearly shown that MOM1 ChIP peaks overlap with MORC peaks, which were distinct from PolV peaks, the DE TEs in *mom1* in Sup Fig6 still formed different clusters from *morc* mutants. Any explanation for the distinct TE targets in the mutants? Do these DE TEs associate with DNA methylation changes in *mom1* and *morc* mutants?

Dear Dr. An. We would like to thank the reviewers for the helpful and constructive comments on our manuscript. Below we respond point-by-point with **reviewer comments in bold** and our comments in regular type.

REVIEWER COMMENTS

Reviewer #1 (Remarks to the Author):

In this manuscript entitled “The MOM1 complex recruits the RdDM machinery via MORC6 to establish de novo DNA methylation”, the authors explored the function of MOM1 complex in establishing and maintaining DNA methylation in Arabidopsis. To explore the function of MOM1, the authors performed ChIP-seq of MOM1 and MOM1 complex components. Unexpectedly, these proteins were all colocalized with Pol V at RdDM sites, suggesting the MOM1 complex may also function in DNA methylation and gene silencing. Using the well-established *fwa* silencing system, the authors demonstrated that tethering of MOM1 complex components to the FWA promoter in the *fwa* mutant by ZF fusion led to the establishment of DNA methylation and silencing of the FWA gene that depends on the Pol V arm of the RdDM pathway and MORC6. The MOM1 complex is also required to maintain a small number of endogenous RdDM target loci. The authors also demonstrated that PIAL2 could interact with MORC6. Based on the genetic results, they proposed that the MOM1 complex recruits the RdDM machinery via MOR6 to establish de novo DNA methylation. Furthermore, the authors identified a group of MOM1-specific targets at active chromatin regions, although the function of MOM1 in these regions is still unclear. In summary, this study reported a new function of MOM1 in DNA methylation and helped explore the role of MOM1 in gene silencing. Overall, this manuscript was well-organized and easy to follow.

Thank you for these positive comments!

Comments:

1. Please perform statistical analysis for flowering time in Fig.2a, Fig. 3a, Fig. 4a and 4c et al.

Thanks for this suggestion. We now performed statistical analysis for all flowering time data and to save space in the figures, the detailed statistics as well as all detailed flowering time and plant numbers for each population are listed in Supplementary Table 3. Please note that in some T1 populations of the ZF tethering assay we observed that the mean flowering time was slightly but statistically earlier than that in the *fwa* control population, while none of the T1 plants exhibited early flowering meeting our threshold (< 20 true leaves). We suspect that this is due to factors other than *FWA* silencing, such as stress in the T1 plants during the resistant marker selection process, replanting of the T1 plants in soil of desired distance and so on.

2. To better support the conclusion, the authors should perform MORC6 ChIP-seq in the *mom1* background.

This is a great suggestion. We demonstrated that ZF tethering of MOM1 recruits the RdDM machinery to the *FWA* promoter via MORC6. To test if a similar recruitment process is also occurring at endogenous RdDM sites, we constructed MYC-tagged MORC6 transgenic lines in the backgrounds of Col-0, *morc6-3* mutant, *mom1-3* mutant and *pial1/2* double mutant and performed ChIP-seq. We observed that the MORC6 ChIP-seq signal over the Pol V peaks were strongly decreased in the *mom1-3* and *pial1/2* mutant backgrounds (Supplementary Fig. 6a and b). The decrease of the MORC6 ChIP-seq signal in the *mom1-3* and *pial1/2* mutant backgrounds was not due to lower MORC6-myc protein level, as suggested by the western blot comparing the MORC6-Myc protein levels (Supplementary Fig. 6c). Thus, MOM1 also facilitates loading of MORC6 protein to endogenous RdDM sites. At the same time, there is still residual MORC6-Myc ChIP-seq signal over the Pol V peaks, suggesting that MORC6 is also recruited to RdDM sites by other mechanisms in addition to the MOM1 complex. This is consistent with previous reports of interactions between MORC6 and DMS3 (Lorkovic et al., Current Biology, 2012) and MORC6 and SUVH9 (Liu et al., PLoS Genetics, 2014; Jing et al., *Molecular Plant*, 2016), which could also recruit MORC6 to RdDM sites.

We have updated the manuscript text correspondingly, by adding the following paragraph: “To investigate if the MOM1 complex also recruits the MORC6 protein at other loci, ChIP-seq was performed with Myc-tagged MORC6 in the backgrounds of Col-0, *morc6-3* mutant, *mom1-3* mutant and *pial1/2* double mutant. MORC6 ChIP-seq signal over Pol V peaks was strongly decreased in the *mom1-3* and *pial1/2* mutant backgrounds compared to that in the backgrounds of Col-0 and *morc6-3* mutant (Supplementary Fig. 6a and b), while the MORC6-Myc protein expression levels were not decreased in the *mom1-3* and *pial1/2* mutant backgrounds (Supplementary Fig. 6c). At the same time, there was still residue MORC6 ChIP-seq signal over Pol V peaks in the *mom1-3* and *pial1/2* mutant backgrounds (Supplementary Fig. 6a and b). These data suggest that MORC6 is recruited to RdDM sites by the MOM1 complex as well as other mechanisms.” (starting from line 275) We also added a corresponding sentence in the discussion section: “We also found that MORC6 ChIP-seq signal at RdDM sites was strongly decreased in the *mom1* and *pial1/2* mutants, suggesting that MORC6 is loaded onto endogenous RdDM sites by the MOM1 complex, together with other mechanisms.” (starting from line 475)

3. In Fig. 5d, the authors performed ATAC-seq in *mom1* and WT and identified 342 regions with increased ATAC-seq signal in the *mom1-3* mutant background. Do these 342 regions overlap with MOM1-binding sites? How many sites are overlapped between 342 regions with CHH-hypo sites?

Thank you for these points. We plotted the MOM1 ChIP-seq signal over the regions with increased ATAC-seq signal in the *mom1-3* mutant, and observed that the MOM1 ChIP-seq signal is significantly enriched over these regions. We updated Figure 5d and the manuscript text correspondingly by adding the following sentence: “As expected, these regions were enriched for MOM1 ChIP-seq signal (Fig 5d).” (line 386). This is consistent with the observation that these regions are enriched for Pol V ChIP-seq signal. However, we found that only 32 of the ATAC-seq increased regions overlapped with hypo CHH hcDMRs in the *mom1-3* mutant. We suspect that the relatively low overlap is due to stringent thresholds for identifying hcDMRs and regions with increased ATAC-seq signal. Thus, we directly examined the DNA methylation level over

the 342 regions with increased ATAC-seq signal in the *mom1-3* mutant and observed that DNA methylation levels in CG, CHG and CHH contexts are all decreased over the majority of these regions. The plot is in Supplementary Fig. 11 and the text in manuscript is updated correspondingly, by adding the following sentence: “Consistently, DNA methylation levels in CG, CHG and CHH contexts were decreased over the majority of these regions (Supplementary Fig. 11).” (line 389).

4. In Fig.2d, please include the Col-0 and *fwa* controls.

Thank you for this suggestion. Actually, Figure 2d is a metaplot of the change of DNA methylation level at the ZF off target sites which was calculated by subtracting the DNA methylation level of the *fwa* mutant from the DNA methylation level of MOM1-ZF, MOM2-ZF, PIAL1-ZF, PIAL2-ZF and PHD1-ZF in the *fwa* background at the ZF off-target sites. We have updated the y-axis label to methylation level variation (sample minus *fwa* control) and also indicated this detail in the figure legend. Also, we have included the metaplot of the original DNA methylation level at the ZF off-target site of MOM1-ZF, MOM2-ZF, PIAL1-ZF, PIAL2-ZF and PHD1-ZF in the *fwa* background, as well as that of Col-0 (GSM2124018) control and the *fwa* control in the Supplementary Fig. 3a.

5. A working model should be proposed to explain the function of MOM1 easily.

A working model illustrating that MOM1 recruits the Pol V arm of the RdDM machinery via MORC6 is now added to Figure 7 and the following sentences have been added to the discussion section: “Based on these observations, we propose a model (Fig. 7) for how the MOM1 complex influences the RdDM machinery as follows: The MOM1 complex is first loaded onto RdDM target sites through an unknown mechanism to facilitate the binding of the MORC6 protein. MORC6 would then enhance the recruitment of the Pol V arm of the RdDM machinery to methylate target DNA, by topologically entrapping the DNA as well as directly interacting with RdDM components, thus serving as a tethering factor.” (starting from line 478).

Reviewer #2 (Remarks to the Author):

Mechanisms that MOM1 affects gene silencing remain enigmatic since the discovery of this gene. Here the authors show that MOM1 functions upstream of MORC and affects silencing and DNA methylation in RdDM targets. That is a new aspect of gene silencing mechanisms and very nice extension of the results of the Jacobsen group on MORC and RdDM. The effects of factors in this new pathway have been extensively studied in this paper. The results are novel and look convincing overall. I have only minor suggestions.

Thank you for these positive comments!

1) It seems that the MOM1 complex has at least two functions, RdDM in euchromatic regions and transcriptional silencing of heterochromatic regions. I wonder if these two functions are connected or not. For example, as the authors mentioned briefly in Discussion for the FWA locus, is it possible that the *mom1* mutation compromises RdDM genome-wide via transcriptional derepression?

We are also fascinated that MOM1 appears to have two functions. However, this is still a puzzle to us. With regard to the referees' proposal that MOM1 plays a universal role in repression, and that the methylation losses at RdDM sites are a secondary consequence of this, we think this is likely not the case because there is not much of a correlation between expression and methylation loss at RdDM sites. For example, in Figure 5c, the transcriptional derepression is very mild in the *mom1* mutant near the hypo CHH hcDMRs in *morchex* mutant (which are RdDM sites and also show lower DNA methylation level in *mom1* mutant). By looking at browser, we can also find many examples of mild methylation loss with no changes in expression. We have removed that point about *FWA* from the discussion because we realize we have no basis for proposing this. In fact, we know that ZF-MOM1 has no silencing effect on *FWA* in *rddm* mutants. (The removed sentence is: "It is also possible that MOM1 complex mutants show defective transcriptional silencing of *FWA* during the DNA methylation establishment process, such that positive epigenetic marks associated with transcription may compete with the methylation establishment process, making it slower or less efficient.")

Given the direct recruitment relationship we observed using the ZF108 / *fwa* *epi-mutant* system between the MOM1 complex, MORC6 and the Pol V arm of the RdDM machinery, we think that the effect of MOM1 complex on endogenous RdDM function is likely by directly influencing the RdDM machinery recruitment, rather than indirectly through transcriptional derepression. We added the following paragraph in the discussion about the relationship between MOM1 function at RdDM and at silencing pericentromeric TEs to draw attention to this interesting open question: "From previous studies and data from this study, it seems clear that the MOM1 complex has at least two functions in epigenome regulation, a role in the establishment and maintenance of RdDM, and a role in the maintenance of silencing of TEs in pericentromeric regions. Furthermore, it appears that these two functions are mechanistically distinct. For example, comparison of DE-TEs and DE-genes in the *nrpe1* and *mom1* mutants in previous studies indicates that the majority of their endogenous targets do not overlap. Thus, for instance, the loss of RdDM function in the *nrpe1* mutant does not impair the silencing function of MOM1 at the majority of its TE targets. It's possible that the localization of MOM1 at RdDM is only needed for the silencing of the relatively small number of shared TE targets between MOM1 and RdDM. In future studies, it will be interesting to investigate the relationship between the two functions of the MOM1 complex, and identify MOM1 complex component(s) or protein domain(s) that might be required for only one of the functions." (starting from line 484).

2) For the same reason, I wonder if the authors can detect differential effects of modified proteins of MOM1 complex for these two pathways. As the authors regard MOM1 has "multifaceted roles" (the last sentence of the Abstract) it would be nice if they can detect region(s) necessary for one of these roles but not for the other.

While this is an interesting suggestion, this would unfortunately take quite a lot of time to do these experiments since we would be starting from scratch with all of the assays, including several plant generations. Also, it is hard to know where to start because currently there is no MOM1 complex component which appears to be important for only one of these two functions of the MOM1 complex. For example, the RdDM function is similarly impaired in *mom1* and *pial1/2* mutants (loss of DNA methylation at endogenous RdDM sites, slowing down of transgene silencing), while only marginally affected in the mutants of other MOM1 complex components (MOM2, AIPP3 and PHD1). At the same time, the pericentromeric TEs are very similarly upregulated in the *mom1* and *pial1/2* mutants, but barely upregulated in *mom2*, *aipp3* and *phd1* mutants. Thus, MOM1 and PIAL1/2 are the key factors needed for both the function of RdDM maintenance and the silencing of pericentromeric TEs.

We have added this interesting point to the discussion and would like to leave this question for future research. The sentence added is “In future studies, it will be interesting to investigate the relationship between the two functions of the MOM1 complex, and identify MOM1 complex component(s) or protein domain(s) required for only one of the functions.” (starting from line 493)

3) It seems that the MOM1 pathway and SUVH2/9 pathway may induce RdDM in parallel. I wonder how epigenome dynamics changes in the *mom1 suvh2/9* triple mutant.

This is an interesting question. While we do not have the *mom1 suvh2/9 triple*, we can provide an analysis of the individual mutants that sheds some light on this question.

In the Zinc Finger 108 tethering assay, SUVH2, SUVH9 (Johnson *et al.*, Nature, 2014; Gallego-Bartolome *et al.*, Cell, 2019) and the MOM1 complex are all able to trigger DNA methylation at the *FWA* gene promoter in the *fwa-4* epi-mutant via recruiting the Pol V arm of RdDM. Thus, endogenously, SUVH2/9 and MOM1 may induce RdDM in parallel at different subsets of RdDM loci, or, it's possible that they have overlapping target loci and reinforce the recruitment of the RdDM machinery by each other to exert a full level of methylation at the target sites. To explore these possibilities, we performed new whole genome bisulfite sequencing of wild type (Col-0), *mom1-3* mutant and *suvh2/9* double mutant plants grown side by side. Consistent with a previous report (Johnson *et al.*, Nature, 2014), CHH DNA methylation is lost in the *suvh2/9* double mutant over the majority of hypoCHH hcDMRs in the *nrpe1* mutant (*nrpe1* DNA methylation data from Hume *et al.*, Cell, 2013) (Supplementary Fig. 10a), suggesting that the recruitment of the RdDM pathway by SUVH2/9 plays a predominant role at most of endogenous RdDM sites.

Interestingly, 46 out of the 120 hypoCHH hcDMRs of the *mom1-3* mutant (shared by two replicates) were not identified as hypoCHH hcDMRs in the *suvh2/9* double mutant. The CHH DNA methylation over these sites was largely preserved in the *suvh2/9* double mutant background (Supplementary Fig. 10b and c), suggesting that MOM1 can still trigger RdDM without SUVH2/9 at these sites. At the same time, over the other 74 *mom1-3* hypo CHH hcDMRs (also identified as hypo CHH hcDMRs in *suvh2/9*), CHH DNA methylation is strongly decreased in the *mom1-3* mutant and in the *suvh2/9* double mutant (Supplementary Fig. 10d), suggesting that MOM1 and SUVH2/9 are both required for RdDM function at these loci. It is possible that at these loci, once an initial low level of DNA methylation is established through

MOM1 mediated RdDM recruitment, SUVH2/9 binds to the DNA methylation and further reinforces the recruitment of the RdDM machinery. In addition, although SUVH2/9 is not required for the silencing of *FWA* by the MOM1-ZF108 tethering assay, we cannot exclude the possibility that the MOM1 complex can somehow recruit SUVH2/9. For example, interaction between MORC6 and SUVH9 has been reported before (Liu et al., PLoS Genetics, 2014; Jing et al., *Molecular Plant*, 2016). Thus, it's possible that the MOM1 complex might also be able to recruit SUVH9 via MORC6.

In addition to the new figures, we have added corresponding text to line 370.

Reviewer #3 (Remarks to the Author):

The manuscript entitled "The MOM1 complex recruits the RdDM machinery via MORC6 to establish de novo DNA methylation" by Li et al. reports a link between the MOM1 complex and RdDM machinery, as well as MORC proteins in Arabidopsis. The authors performed Mass-spec analysis to identify proteins interacting with MOM1 and identified core proteins, including MOM1/2, PIAL1/2, PHD1, and AIPP3. The authors further examined the role of the MOM1 complex in de novo DNA methylation and epistasis with RdDM and MORC genes using the *FWA* silencing assay, demonstrating the involvement of the MOM1 complex in RdDM-dependent silencing of the *FWA* locus. In addition, they showed MOM1 localizations independent of RdDM target loci.

Overall, the experiments performed in this work were well designed, and conclusions are very clear and supported by the rigorous results, including those with MASS-spec and transgenic assays. The findings, including the role of MOM1, which was long thought as independent of DNA methylation, in RdDM and MORC pathways are novel and interesting, and would have a broad impact in the field. I would recommend this manuscript for publication in Nature Communications, provided the authors address the following points.

Thank you for these positive comments!

-Figure 3a: It looks like the introduction of MOM1-ZF to *drm1/2 fwa* still induced early flowering in some transgenic T1 plants. Is there any possibility of RdDM independent silencing mechanisms of *FWA* gene such as H3K9me or PTGS in this system?

Thank you for this insightful question! We also noticed that some of the MOM1-ZF T1 plants in *drm1/2 x fwa* mutant background showed relatively early flowering time compared to the *fwa* control (although none of the 59 T1 plants had < 20 true leaves that met our threshold). If this relative early flowering phenotype is due to RdDM independent silencing of the *FWA* gene by MOM1, we should be able to see this by looking at gene expression. Therefore, we collected rosette leaves from six T1 plants which showed the earliest flowering time within this T1

population (with true leaf number from 23 to 25) and quantified the expression level of the *FWA* gene by qRT-PCR. The *FWA* gene expression level was not decreased in these plants, compared to the *fwa-4* control (Supplementary Fig. 5), suggesting that the relatively early flowering time was due to other factors, such as stress during the T1 selection. Correspondingly, we added the following sentences to line 243: “Some of MOM1-ZF T1 plants displayed intermediate flowering time (20-30 true leaves) in *drm1/2 fwa*, *dms3 fwa*, *drd1 fwa* and *rdm1 fwa* backgrounds (Fig. 3a). However, *FWA* gene expression was not decreased in the six MOM1-ZF T1 plants in the *drm1/2 fwa* background which had the earliest flowering time (23-25 true leaves) from this population (Supplementary Fig. 5), suggesting that the intermediate flowering phenotype is likely due to other factors such as stress rather than *FWA* silencing.”

To further check if MOM1-ZF mediates RdDM independent silencing at the *FWA* gene, we performed RNA-seq of MOM1-ZF T2 plants in the *nrpe1 fwa* background. MOM1-ZF protein signal was detected in all four T2 populations used (Picture below, panel **a**). RNA-seq of these T2 populations revealed that expression level of the *FWA* gene was not decreased compared to the *nrpe1 x fwa* control plants (Picture below, panel **b**), suggesting that MOM1-ZF was not able to silence the *FWA* gene in the *nrpe1 x fwa* background. Overall, our data suggests that the establishment of silencing at the *FWA* gene by MOM1-ZF is solely dependent on the recruitment of the RdDM machinery. Although it seems clear that that the MOM1 complex has divergent endogenous function separate from that of the RdDM machinery, it’s possible that the silencing by MOM1 requires some level of DNA methylation. If this is true, without the establishment of DNA methylation by the RdDM machinery, the MOM1 complex alone would not be able to exerts its silencing function. To save text and figure space of the manuscript, we have not included the result of MOM1-ZF T2 plants in *nrpe1 x fwa* in the manuscript as it’s another piece of negative data suggesting same conclusion as Supplementary Fig. 5.

***FWA* gene was not silenced by MOM1-ZF in *nrpe1 x fwa* background.** **a**, Western blot of four T2 populations of MOM1-ZF in *nrpe1 x fwa* background. M2 antibody was used to detect the flag epitope of the MOM1-ZF protein. Eight rosette leaves from different T2 plants of the same T2 population were pool harvested for protein extraction. *nrpe1 x fwa* leaves were used as negative control. Ponceau S staining of transferred membrane was used as loading control. **b**, *FWA* expression level (TPM) of MOM1-ZF T2 plants in *nrpe1 x fwa* background. *nrpe1 x fwa* were used as control. Rosette leaves of four transgene positive plants from each T2 population were used for RNA-seq. Bar plots and error bars indicate the mean and standard error of four replicates, respectively, with individual technical replicates shown as dots.

-Line 326-333: Even though the description of overlaps of CHH DMRs between *morchex* and *mom1*, *pail1/2*, *aipp3* etc., no data was shown for the exact number of the overlapping DMRs. Sup. Table 3 shows only the total numbers and positions of DMRs. The number of overlapping DMRs or Venn diagram can be provided.

Thanks for this suggestion. We now provide the exact number of overlapping hypo CHH hcDMRs as well as the total number of hypo CHH hcDMRs of each mutant in Fig. 5b and the negative natural log of P-value for this overlap using the hypergeometric distribution calculated by *homer mergePeaks* in Supplementary Fig 9. The manuscript at line 354 is updated correspondingly.

-Line 366-367; “We also discovered a set of MOM1 ChIP peaks that did not overlap with DNA methylation”. I think the data of DNA methylation levels associated with the MOM1 ChIP peaks were not shown in this manuscript.

Thank you for pointing this out. We plotted DNA methylation level over Cluster 1 peaks (MOM1 and Pol V co-binding peaks) and Cluster 2 peaks (MOM1 unique peaks), and observed that as expected, Cluster 1 peaks are DNA methylated in all three sequence contexts (CG, CHG and CHH), while DNA methylation level at Cluster 2 peaks (MOM1 unique peaks) is very low. The corresponding plot is in Supplementary Fig. 12c and the manuscript text is updated correspondingly, by adding the following sentence at line 418: “As expected, the Cluster 1 peaks were DNA methylated in all sequence contexts, while DNA methylation levels over Cluster 2 peaks were very low (Supplementary Fig. 12c)”

-Figure 5a: Although the manuscript mainly focused on CHH methylation, *mom1* and *pail1/2* showed variations in CGm and CHGm in quite some DMRs. Are these DMRs associated with TEs in pericentromeric regions?

Thanks for this great question. We now systematically analyzed the hypo CHH, hypo CHG and hypo CHG hcDMRs in *mom1-3* and *pial1/2* mutants with our new WGBS data. We observed 120 hypo CHH hcDMRs in *mom1-3* (shared by two replicates), and 93 hypo CHH hcDMRs in *pial1/2* double mutant. Over these hypo CHH hcDMRs, Pol V ChIP-seq signal is enriched and the CHH methylation level are also strongly decreased in the *nrpe1* mutant, suggesting that these hypo CHH hcDMRs in *mom1-3* and *pial1/2* mutants are RdDM sites (Supplementary Fig. 8a). Similarly, the hypo CHG hcDMRs in *mom1-3* and *pial/2* mutants are also mainly RdDM sites (Supplementary Fig. 8b). The hypo CHH and hypo CHG hcDMRs are more enriched in the pericentromeric region compared to the chromosome arms (Supplementary Fig. 8e). On the contrary, the majority of hypo CG hcDMRs in *mom1-3* and *pial1/2* mutants are not enriched for Pol V ChIP-seq signal, are devoid of CHH, CHG methylation in Col-0, and are located in genes and in chromosome arms, suggesting that they are likely sites of gene body methylation (Supplementary Fig. 8c-e). Only a very small proportion of hypo CG hcDMRs in *mom1-3* and in *pial1/2* double mutant overlapped (Supplementary Fig. 8f), suggesting that the majority of these hypo CG hcDMRs are unlikely due to the function of the MOM1 complex. It's

likely that these hypo CG hcDMRs are accumulated random natural variations in CG methylation. We updated the manuscript text starting at line 339 correspondingly.

-Figure 6, Line 437-439: While it is clearly shown that MOM1 ChIP peaks overlap with MORC peaks, which were distinct from PolV peaks, the DE TEs in *mom1* in Sup Fig6 still formed different clusters from *morc* mutants. Any explanation for the distinct TE targets in the mutants? Do these DE TEs associate with DNA methylation changes in *mom1* and *morc* mutants?

This is a very interesting question. Although many similarities have been identified between the MOM1 complex and the MORC proteins, there are also differences that could explain the different spectrum of DE TEs in the mutants. For example, it is known that synergistic repression of some TEs and genes by MORC6 and MOM1 can occur, because a comparison of RNA-seq data from *mom1*, *morc6* and *mom1 morc6* double mutant showed a much greater TE derepression in the double mutant (Moissiard et al., PNAS, 2014). In contrast, the derepression of TEs was not enhanced in the *mom1 pial2* double mutant compared to the *mom1* single mutant (Han et al., The Plant Cell, 2016), in line with the PIAL proteins and MOM1 working in the same pathway. Thus, MORC6 and MOM1 appear to work at least partially by different mechanisms and we do not therefore expect the spectrum of upregulated TEs to fully overlap between the mutants of MORC and MOM1 complex mutants. We added the following sentence to line 409: “Meanwhile, many upregulated TEs in the *mom1-3* mutant are not derepressed or only mildly derepressed in the *morc6-3* and *morchex* mutants, suggesting that the functions of the MOM1 complex and the MORC proteins do not fully overlap.”

From this same previous study (Moissiard et al., PNAS, 2014), most upregulated TEs in the *morc6* mutant were also upregulated in the *mom1* mutant, while many TEs upregulated in the *mom1* mutant were not upregulated in the *morc6* mutants. Our heatmap also suggests that the cluster of TEs which are upregulated in the *morc6* and *morchex* mutant are also upregulated in the *mom1* mutant. We compared the DNA methylation level of Col-0 control versus the mutants over the TEs that are upregulated in *morchex* mutant (n = 85, MOM1 and MORC co-regulated TEs), and TEs that are only up-regulated in *mom1-3* (n = 102, MOM1 distinct TEs targets). Although there are some differences in DNA methylation levels between different batches of WGBS data (likely due to well-known technical batch artifacts), within each batch of data, there is minimal DNA methylation level difference between *mom1*, *morc6* and *morchex* mutant compared to their corresponding Col-0 control (see the plot below). These results are consistent with previous studies reporting that there is no major DNA methylation loss over MOM1 targets and MORC targets in the *mom1* mutant and *morc6* mutants, respectively (Han et al., The Plant Cell, 2016; Moissiard et al., PNAS, 2014; Zhang et al., PNAS, 2018; Moissiard et al., Science, 2012). We have not added these data to the manuscript as they are a re-iteration and validation of previously reported observations.

Over MOM1 and MORC co-regulated TEs (n = 85)

Over MOM1 distinct TE targets (n = 102)

DNA methylation levels are similar between the *mom1*, *morc* mutants and the Col-0 control over upregulated TEs. DNA methylation levels in CG, CHG and CHH contexts of (1) the *mom1-3* mutant and corresponding Col-0 control (Data from this study); (2) the *morc6* mutant and corresponding Col-0 control (Data from GSE54677, Moissiard *et al.*, PNAS, 2014); (3) the *morchex* mutant and Col-0 control (Data from GSE78836, Harris *et al.*, Plos Genetics, 2016) are plotted over TEs upregulated in the *morchex* mutant as MOM1 and MORC co-regulated TEs (upper panel) and over TEs upregulated in the *mom1* mutant but not in the *morchex* mutant as MOM1 distinct TE targets (lower panel).

Reviewers' Comments:

Reviewer #1:

Remarks to the Author:

This manuscript has been much improved; the authors have addressed all my concerns. Thanks!

Reviewer #2:

Remarks to the Author:

The manuscript has been improved by incorporating reviewers comments. I do not have any further suggestions to improve the manuscript.

Reviewer #3:

Remarks to the Author:

The authors addressed my concerns and questions with sufficient additional experiments and analysis, which support the conclusions of the manuscript. I have no further comments on the revised manuscript.

Dear Dr. An. We would like to thank the reviewers for their evaluation of our manuscript. Below we respond with **reviewer comments in bold** and our comments in regular type.

REVIEWER COMMENTS

Reviewer #1 (Remarks to the Author):

This manuscript has been much improved; the authors have addressed all my concerns. Thanks!

We thank this reviewer for the comments!

Reviewer #2 (Remarks to the Author):

The manuscript has been improved by incorporating reviewers comments. I do not have any further suggestions to improve the manuscript.

We thank this reviewer for the comments!

Reviewer #3 (Remarks to the Author):

The authors addressed my concerns and questions with sufficient additional experiments and analysis, which support the conclusions of the manuscript. I have no further comments on the revised manuscript.

We thank this reviewer for the comments!